# Integrated Management of Pathogens and Microbes in *Cannabis sativa* L. (Cannabis) under Greenhouse Conditions

**DOI:** 10.3390/plants13060786

**Published:** 2024-03-10

**Authors:** Liam Buirs, Zamir K. Punja

**Affiliations:** 1Pure Sunfarms Corp., Delta, BC V4K 3N3, Canada; lbuirs@puresunfarms.com; 2Department of Biological Sciences, Simon Fraser University, Burnaby, BC V5A 1S6, Canada

**Keywords:** biological control, bud rot, cultural control, fungal diseases, plant pathogens, root rots

## Abstract

The increased cultivation of high THC-containing *Cannabis sativa* L. (cannabis), particularly in greenhouses, has resulted in a greater incidence of diseases and molds that can negatively affect the growth and quality of the crop. Among them, the most important diseases are root rots (*Fusarium* and *Pythium* spp.), bud rot (*Botrytis cinerea*), powdery mildew (*Golovinomyces ambrosiae*), cannabis stunt disease (caused by hop latent viroid), and a range of microbes that reduce post-harvest quality. An integrated management approach to reduce the impact of these diseases/microbes requires combining different approaches that target the reproduction, spread, and survival of the associated pathogens, many of which can occur on the same plant simultaneously. These approaches will be discussed in the context of developing an integrated plan to manage the important pathogens of greenhouse-grown cannabis at different stages of plant development. These stages include the maintenance of stock plants, propagation through cuttings, vegetative growth of plants, and flowering. The cultivation of cannabis genotypes with tolerance or resistance to various pathogens is a very important approach, as well as the maintenance of pathogen-free stock plants. When combined with cultural approaches (sanitation, management of irrigation, and monitoring for diseases) and environmental approaches (greenhouse climate modification), a significant reduction in pathogen development and spread can be achieved. The use of preventive applications of microbial biological control agents and reduced-risk biorational products can also reduce disease development at all stages of production in jurisdictions where they are registered for use. The combined use of promising strategies for integrated disease management in cannabis plants during greenhouse production will be reviewed. Future areas for research are identified.

## 1. Introduction

Integrated disease management (IDM) incorporates the coordinated use of multiple approaches to reduce the impact of disease-causing agents (pathogens) on agricultural crops [1]. When applied in parallel or consecutively, these tactics can achieve control of multiple pathogens using different and sometimes synergistic suppression tactics. IDM builds upon the concept of integrated pest management (IPM), which has been widely utilized for decades to target and manage insect pests on agricultural crops and requires different strategies to be employed in a coordinated manner, often with resounding success [2,3]. When IDM approaches are considered for cannabis (*Cannabis sativa* L., high THC-containing genotypes) grown under greenhouse conditions, several aspects need to be modified from traditional IDM programs. First and foremost is the fact that there are no synthetic fungicides available for use on cannabis crops, thus eliminating a widely-used disease management strategy. Instead, only reduced-risk “biological” and “biorational” products are permitted. These products are mostly protective in action, i.e., non-fungicidal, so they are best suited for preventative applications, although some products can also be deployed as sanitizers. While claims of product efficacy and applications for disease reduction in cannabis may be made, not all are necessarily supported by data from replicated research trials or third-party evaluations. This adds to the difficulty in identifying the specific IDM approaches that are best suited for each pathogen. The recent expansion of hemp cultivation (*C. sativa*, low THC-containing cultivars) in the USA following federal government approval should provide useful information on disease and pest management approaches that could be extended to cannabis [4]. The lack of synthetic fungicides for cannabis production has prompted the registration of several biological control products that can be used at different stages of production [5,6]. However, efficacy data for these products are not always available, and the modes of action of the biocontrol agents are not often fully understood in the context of cannabis IDM, highlighting the need for further research in this area [6,7]. Fortunately, efficacy data may exist for many of these products on other crops, e.g., for organic production, and therefore, IDM approaches utilized in these crops can likely be extrapolated to cannabis crops [8]. A second challenge for IDM development in cannabis is that highly bred cultivars containing specific resistance genes against important pathogens are lacking. Instead, genetic selections (genotypes) that target higher yields of inflorescences and THC content and that display unique morphological traits have been made a priority [9]. In most instances, these efforts have excluded the specific incorporation of disease resistance traits. Consequently, some high-yielding genotypes frequently show high susceptibility to various pathogens, as will be illustrated in this review. Fortunately, the broad genetic variation that currently exists among cannabis genotypes has led to the identification of resistance in various genotypes to specific pathogens, such as powdery mildew [6,10,11,12]. The mechanisms underlying this resistance are currently under investigation [13].

A third challenge is that when cannabis is compared to other widely-grown greenhouse crops, such as tomatoes, cucumbers, and peppers, the optimal cultural and environmental conditions for cultivation have not yet been fully established. Since different cannabis greenhouse operations can experience variable growing conditions, standardized research trials are needed to establish these parameters. Recent research has identified integral aspects of controlled environment cultivation practices that can be used as a baseline reference [14,15]. The prevalent pathogens affecting cannabis crops in greenhouses have been recently characterized and described [7], providing diagnostic information that is required for IDM implementation. Accurate diagnosis of the pathogen(s) involved in a disease syndrome is an important component of IDM, and several diagnostic methods have been described [4,7,16,17,18,19]. In this review article, we describe the most important pathogens of cannabis crops cultivated under greenhouse conditions and highlight the various growth stages at which IDM approaches can be implemented during the crop production cycle, which generally occurs over 12–15 weeks (Figure 1).

The first stage of production of a cannabis crop is the cultivation and maintenance of stock (mother) plants (Figure 2a), which provide a source of vegetative cuttings (Figure 2b). Once cuttings are rooted, they are transferred to greenhouse growing conditions for 2–3 weeks to continue vegetative growth (Figure 2c). The developing vegetative plants are then transferred to flowering rooms for 7–8 weeks (Figure 2d,e), after which time the inflorescences are harvested (Figure 2f).

During each crop production year, up to 3–4 cropping cycles may take place per greenhouse compartment. The IDM approaches that can be developed include selection of disease-tolerant genotypes, implementation of cultural practices, modification of environmental climate settings, and application of reduced-risk products (Figure 3).

We also discuss aspects of the microbial colonization of cannabis inflorescences by yeasts and molds and propose IDM strategies to reduce the total microflora present. Monitoring the microbial colonization of inflorescences is an important quality aspect of cannabis, which is under strict regulatory control and presents a unique and challenging component of crop management that is not found in most other crops [19,20]. This review should aid in the design or refinement of further IDM programs in greenhouse-cultivated cannabis operations. Detailed descriptions of the symptoms caused by various pathogens at different stages of cannabis growth during commercial production and the approaches that can be taken to manage them are described below.

## 2. Cannabis Pathogens: Symptoms and Management Approaches at Different Stages of Growth

### 2.1. Stock Cultivation Stage

Stock (mother) plants provide a source of vegetative cuttings, which are commonly used in commercial cannabis production. These plants generally constitute a range of genotypes that are chosen for their desired phenotypic characteristics and biochemical profiles. They are grown in designated areas within the greenhouse or in separate indoor rooms. Physical separation of stock plants from larger-scale commercial production is important to prevent the spread of pathogens. The ages of these stock plants can vary and typically range from 3 to 12 months, depending on the facility. In the context of disease development, older plants often exhibit signs of declining growth, such as reduced shoot growth, leaf yellowing, and poor root development (Figure 4a). These symptoms may be indicative of sub-lethal infections by *Fusarium* and *Pythium* spp. or hop latent viroid (Figure 4 and Figure 5).

A closer inspection of the stems of diseased plants will often reveal internal discolouration in the pith and xylem tissues (Figure 4b,c), a symptom of *Fusarium* infection, and/or root browning that can be caused by *Fusarium* or *Pythium* species [21,22,23,24]. Excessive waterlogging may also cause root browning on cannabis plants. Accurate pathogen diagnosis at this stage is critical to determine the most effective IDM strategies to implement. Stock plants are also susceptible to powdery mildew, which is clearly visible as white colonies on the upper surfaces of leaves (Figure 4g). A significant challenge in maintaining healthy stock plants is the recent emergence of hop latent viroid (HLVd) [25,26,27], which is mostly asymptomatic on stock plants but may cause occasional curling or mottling on the youngest leaves (Figure 4h,i). The impact of HLVd infection in stock plants is seen when rooting frequency and vigor of cuttings derived from them are examined (Figure 5). HLVd infection leads to poor root growth (Figure 5b) that continues to impact plant growth at the vegetative stage (Figure 5c,d) and can also impact flowering (Figure 5e). HLVd-infected flowering plants derived from infected stock plants display reduced inflorescence growth as well as lower levels of cannabinoid production [26]. This emphasizes the importance of maintaining pathogen-free stock plants during commercial production. Routine scouting for the presence of disease symptoms and testing stock plants for the presence of HLVd, *Fusarium,* and *Pythium* species is highly recommended.

### 2.2. IDM Approaches at the Stock Cultivation Stage

During the stock plant cultivation stage, various IDM strategies can be implemented to minimize the development of plant pathogens. The following are examples of some commonly used practices.

#### 2.2.1. Biosecurity and Quarantine Inspection

Biosecurity practices, which include foot baths, wearing protective clothing, and removing pruned leaves and diseased plants, are standard in most horticultural greenhouse operations [28]; these practices should be implemented for cannabis growing operations. In addition, it is important to establish a quarantine protocol in cases where plant materials, such as unrooted cuttings or whole plants, are brought in from an external source [2]. Such precautionary measures can prevent pathogen introduction and are standard biosecurity protocols in commercial crop production [29]. When applied to cannabis, this necessitates an isolation period of 3–4 weeks, during which plants are monitored for disease symptoms and tested for the presence of potential viruses and other pathogens [6]. Testing for cannabis pathogens can be achieved by polymerase chain reaction (PCR) methodologies, and testing for viruses or viroids can be achieved using reverse transcription polymerase chain reaction (RT-PCR); many laboratories currently offer this service for an array of cannabis pathogens [4,6,7]. After the plants are confirmed to be free of detectable pathogens, they can be used for commercial propagation. Infected plants should be destroyed.

#### 2.2.2. Cultural and Environmental Management

Environmental management is a component of IDM across all stages of cannabis growth since climatic conditions can influence both plant growth as well as pathogen growth. Standard cannabis cultivation environmental setpoints, which are established for baseline pathogen management during low disease pressure periods, have been described [15,30]. Conditions that are unfavorable for disease development while at the same time supporting optimal plant growth are required. This often necessitates lowering temperature and humidity levels below the optimal set points for plant development in order to reduce the vapour pressure deficit (VPD) in the crop environment during periods of high disease pressure. Optimal environmental conditions vary across the different developmental stages of cannabis. In the cloning or seedling stage, temperatures should be kept between 20 °C and 24 °C with relative humidity above 90%. For the stock plant and vegetative stages, temperatures should range from 25 °C to 28 °C to promote rapid growth, with relative humidity maintained between 65% and 75% (equating to a VPD of 1.1 to 0.94). During the flowering stage, temperatures should be set between 23 °C and 28 °C to facilitate the transfer of photosynthates to the flowers, with relative humidity between 50% and 70% (resulting in a VPD of 1.4 to 1.13). These environmental parameters can be achieved by modifying venting, heating, and air circulation strategies [30]. Seasonal adjustments may also need to be made, as warmer temperatures with higher humidity in the summer months may increase the incidence of root-infecting pathogens, such as *Fusarium* and *Pythium* species. Similarly, cooler and more humid conditions during winter seasons may enhance the development of powdery mildew infections. The impact of environmental conditions on HLVd development is currently unknown.

#### 2.2.3. Sanitary Practices

Thorough sanitation of the growing environment before planting a new cannabis crop is important to reduce residual pathogen inoculum, which can be spread by water or air or on tools and potentially on clothing, gloves, or shoes. This is a common practice used on most greenhouse crops, especially where viruses are of concern [6,28]. To reduce pathogen transmission, all surfaces and equipment, as well as gutters, tables, floors, drip emitters, and pots, should be cleaned by using reduced-risk sanitary products. These products include hydrogen peroxide with peracetic acid (Sanidate^®^ or Zerotol^®^), dodecyl dimethyl ammonium chloride (Chemprocide^®^ or KleenGrow^®^), isopropyl alcohol, and bleach [6]. The efficacy of these products in inhibiting pathogen growth can vary depending on the pathogen, product, and concentrations used. A comparison of two products used at four concentrations against the growth of two pathogens is shown in Figure 6a,b. At increasing concentrations, both Zerotol^®^ and hypochlorous acid (a product containing 1000 ppm that was diluted) reduced pathogen growth, but *Pythium* showed a greater sensitivity compared to *Fusarium* (Figure 6c). These products can also potentially negatively affect the growth of beneficial *Trichoderma* species applied as biocontrol agents (Figure 6d). Therefore, care must be taken to consider the potential impact of applying reduced-risk products in conjunction with biocontrol products. These types of evaluations are important to conduct for any reduced-risk product targeted for the cannabis market to demonstrate efficacy and possible non-target effects.

#### 2.2.4. Testing for Pathogen Presence and Eradication

Early detection of disease symptoms in stock plants is important to prevent pathogens from spreading within the growing environment. There are several diagnostic approaches that have been described to detect cannabis pathogens, and a number of commercial laboratories provide testing services for a range of pathogens [4,7,19,26]. The practice of culling and replenishing stock plants is a standard component of IDM programs when diseased plants are detected [31]. Stock plants should be replaced after several (3–4) months of production with new, pathogen-free plants, which is key to maintaining a healthy and vigorous stock plant population. Plants infected with *Fusarium*, *Pythium,* or HLVd should be promptly removed from a facility. Eradication of diseased plants, particularly those infected with HLVd, is essential. When regular (weekly) pathogen testing is followed by the destruction of those plants infected by HLVd, a gradual decline in the occurrence of diseased stock plants can be achieved (Figure 7). After many rounds of testing performed over a 6-month period, this strategy was shown to reduce HLVd frequency in stock plants from 22% to 1% (Figure 7). Peaks of infection can still be seen, which are attributed to the re-introduction of diseased plant material that went undetected initially and was inadvertently used as a source of cuttings. Removing this material upon detection resulted in the continued downward trend of infection.

#### 2.2.5. Utilizing Disease-Tolerant Genotypes

The utility of disease-tolerant genotypes that may have been developed through selective breeding and genotype screening is an important aspect of IDM for stock plants. Disease-tolerant genotypes of cannabis have been identified for a number of pathogens, including root rot (*Fusarium oxysporum*) [23], powdery mildew (*Golovinomyces ambrosiae*) [10,11,12,32], leaf blight (*Neofusicoccum parvum*) [33], and bud rot (*B. cinerea*) [34,35] (Figure 8). Recent research suggests that specific defense genes may play a role in certain host–pathogen interactions, leading to a resistant phenotype [11,12,13,36]. The impact of cannabis genotype on disease development at the flowering stage will be discussed later in this review. Continued evaluation of cannabis genotypes for pathogen response is a critical component of an IDM program.

### 2.3. Propagation Stage

Cannabis plants can be initiated from seeds or from vegetative cuttings, which originate from stock plants, but the latter is more commonly used in commercial production, and large-scale propagation from seeds is less common. Routine testing for pathogens that may be present in seeds is not currently a standard practice in the cannabis industry, which can result in the spread of seed-borne pathogens. Cannabis and hemp seeds are known to harbour species of *Alternaria*, *Fusarium*, and several post-harvest molds [33,37], as well as HLVd [26,27]. Implementing stringent sanitation protocols and testing for fungal or bacterial pathogens using PCR and for viruses or viroids using RT-PCR, as described previously, are important IDM approaches during plant propagation in greenhouse crops, including cannabis [4,7,38,39]. These steps can minimize the subsequent spread of fungal, bacterial, and viral/viroid pathogens. Vegetative cuttings used for propagation are required to be rooted under high-humidity conditions over a two-week period. This environment is conducive to the spread of pathogens such as *Fusarium* spp. and *B. cinerea* (Figure 9), as well as a number of bacteria that may be spread by water or in the air. Testing conducted in the rooting environment by swabbing surfaces, sampling of water and air, or plating of surface-sterilized plant material can be used to assess total microbes that may be present [21,22,23]. Cuttings may unknowingly harbour inoculum of *Fusarium* spp., and infection by powdery mildew or HLVd is likely to be present if the original stock plants were infected [6,7,23,26]. Cuttings taken from stock plants infected internally by *Fusarium* species can result in the spread of the pathogen, resulting in damping-off symptoms, particularly in susceptible genotypes (Figure 9). The infection causes the pith and xylem tissues to collapse, resulting in the death of the cuttings. Powdery mildew symptoms may also appear on cuttings from inoculum either carried over from the stock plants or introduced at the propagation stage. The most significant pathogen affecting root development and growth of cuttings is HLVd, which originates from infected stock plants [26]. Additionally, under high humidity conditions, vegetative cuttings may be affected by gray mold (*B. cinerea*) and common saprophytic fungi, including *Penicillium* spp., which can potentially reduce the appearance and quality of the cuttings [7,23]. Many of these fungal pathogens that affect cannabis cuttings, as well as other stages of plant development, produce large numbers of spores, which can be spread by water, air, and tools throughout the growing facility (Figure 10). These spores can serve as sources of initial inoculum and can be challenging to manage. The inclusion of HEPA filters and HVAC systems is advisable to reduce the total counts of air-borne fungal propagules.

### 2.4. Propagation Stage IDM Approaches

#### 2.4.1. Cultural and Environmental Management

During the propagation stage, ensuring that cuttings are obtained from healthy stock plants reduces the probability of pathogens being transferred through these cuttings. In particular, the incidence of *F. oxysporum* is reported to be greater in cuttings taken from the base of the plant compared to locations higher up the plant [7,23]. Therefore, cuttings from the uppermost part of stock plants may limit transmission of this pathogen and possibly of HLVd, although sufficient data is lacking at the present time for this latter pathogen. Ensuring that cuttings are acclimatized in a transitional environment prior to resuming vegetative growth reduces stress on the rooted plants [38,39].

#### 2.4.2. Application of Biological Control Agents

Several biological control products containing *Trichoderma* spp. or *Gliocladium catenulatum* are registered for use on cannabis in Canada [5]. These products are classified as “reduced risk” and provide an alternative in the absence of registered synthetic fungicides. They can be used at all stages of cannabis crop growth but are particularly useful for managing damping-off caused by *Fusarium* spp. on cuttings. When applied at the vegetative stage of plant growth, they can reduce mortality due to *Fusarium* and *Pythium* species [40]. Several weeks after application, the biocontrol agents can be recovered from cannabis tissues, indicating they are able to survive for a period of time (Figure 11). Their effectiveness is based on the protection of tissues, and therefore, they should be applied before pathogen infection occurs, ideally as a drench or as a dip when cuttings are being rooted or as a drench at later stages of crop growth. The benefits of late applications should be evaluated as most biocontrol products are costly to use at large scales. Biocontrol agents protect susceptible root tissues from infection by root pathogens and can colonize cuttings internally, possibly functioning as endophytes; they can potentially enhance root and shoot growth in addition to providing protection against pathogens [40]. *Trichoderma* spp. also exhibits direct antagonism to *F. oxysporum* in dual culture (Figure 12). However, the optimal conditions for maximizing the efficacy of registered biocontrol agents in cannabis cultivation remain unexplored. Nevertheless, several biocontrol agents have been demonstrated to be effective against root-infecting pathogens when applied preventatively and in accordance with their label on cannabis plants [40], indicating their adaptability to various environmental conditions.

### 2.5. Vegetative Growth Stage

Following the establishment of rooted plants from cuttings, the plants are allowed to continue vegetative growth for an additional 2–3 weeks before being transferred to flowering rooms. During this growth stage, root-infecting pathogens, including *Fusarium* and *Pythium* species, as well as HLVd, may continue to develop and spread. The development of powdery mildew may also become more severe at this stage of production. Internal stem infections by *Fusarium* spp. in rooted cuttings can significantly reduce the growth and development of vegetative plants. Symptoms such as yellowing, stunted growth, browning of roots, and plant death are often linked to infection by *Fusarium* and *Pythium* species (Figure 13). The development of these pathogens can be exacerbated by root damage and excessive watering or flooding, which can also spread the pathogen inoculum and cause further development of disease. Testing of recirculated water for pathogen presence is an important aspect of IDM.

In addition, HLVd infection of rooted cuttings can adversely affect root development and plant growth at the vegetative stage, leading to reduced plant size, particularly in susceptible genotypes (Figure 5). Molecular diagnostic methods should be used to ensure that vegetative plants are not infected by this viroid [4,26]. In greenhouse environments where the recycling of nutrient solutions is practiced, monitoring for the presence of *Fusarium* and *Pythium* inoculum is necessary since both are known to be present in hydroponic nutrient solutions [21]. Regular testing of electrical conductivity (EC) and potential hydrogen (pH), coupled with testing of drip and drain nutrient ratios, will ensure that the nutrient profiles remain within the optimal range for crop development, preventing nutrient deficiencies that could lead to a predisposition to pathogen infection [41,42]. In addition, monitoring water temperature and oxygen levels can reduce extremes that can enhance root infection by pathogens [42]. Treatment of recirculated water with reduced-risk products, such as those indicated in Section 2.2.3, can reduce the incidence of root pathogens (Figure 6). Regular monitoring of plants for symptom development should be conducted.

### 2.6. Vegetative Growth Stage IDM Approaches

#### 2.6.1. Cultural and Environmental Management

Root pathogen development in vegetative plants can be minimized by increasing the interval between watering events, leading to fewer and shorter irrigation events as long as adequate moisture is provided for optimal root development. This strategy has been used to reduce root pathogen development in various crops [2]. On the foliage, the exposure of plants to ultraviolet radiation, especially UV-C light (234 nm wavelength), can suppress powdery mildew mycelium development and spore germination when applied routinely at an appropriate dosage with good coverage of the upper leaf surfaces [43]. Night-time exposure enhances pathogen susceptibility by limiting light-activated DNA repair mechanisms [44]. Exposure to UV-C may also enhance plant defense responses, including the accumulation of reactive oxygen species [45], although the effect on cannabis plants has not been determined. To avoid phytotoxicity, exposure of plants to UV-C should be made gradually over several weeks, according to the manufacturer’s guidelines. Treated plants may show a reduction in plant height and increased lateral branching, as has been observed in some ornamental plant species [46,47]. Additional research is required to demonstrate the potential benefits of UV-C exposure to cannabis plants.

#### 2.6.2. Application of Biological Control Agents

Biological control agents can also be applied as drenches to vegetative plants to reduce the severity of root pathogens, similar to treatments made at the propagation stage [40]. The extent to which these agents can survive following application at this stage has not been determined. Colonization of the rapidly growing roots by the biocontrol agent is required for adequate reduction in pathogen development.

### 2.7. Flowering Stage

After vegetative plants have been transferred to greenhouse compartments where the photoperiod is reduced from 18:6 h light:dark to 12:12 h or other iterations of light:dark [48,49], the onset of inflorescence development is triggered within 1–2 weeks. At this stage of crop development, symptoms of root infection by *Fusarium* or *Pythium* spp. originating from the propagation/vegetative stage may rapidly become apparent. These symptoms include leaf yellowing, plant wilting, crown and root rot, and stunted growth (Figure 14). There is no evidence that new infections from residual inoculum are occurring in flowering plants if all sanitary practices have been followed and recirculated water is not being used. Symptoms attributed to HLVd infection, which may have been previously undetected on vegetative plants, will typically manifest within 1–3 weeks after transfer to the flowering room. These symptoms are distinct, appearing as reduced inflorescence size, yellowing of the bract leaves, and stunted plant growth [26] (Figure 5). The environmental conditions during inflorescence development, which include higher humidity due to increased plant biomass, may also promote the development of powdery mildew, particularly in more susceptible genotypes. Closer to the harvest period, when inflorescences begin to mature, bud rot caused by *B. cinerea* is likely to become visible, depending on environmental conditions and the genotype. This can lead to significant reductions in inflorescence quality and yield (Figure 14). The development of pathogens in cannabis plants during the flowering phase is deemed to have the most significant impact on economic returns and can be the most difficult to manage.

In addition to the above pathogens that infect the crop during the flowering stage, colonization of inflorescences by yeasts and molds prior to harvest is common and generally remains undetected until after harvest when quality tests are performed. On the inflorescence tissues, the most commonly encountered fungal genera include *Penicillium, Alternaria, Cladosporium*, and *Fusarium* (Figure 15). These microbes can be detected by conducting bud swab tests as described in recent studies [19,20,50]. This buildup of yeasts and molds can lead to the final dried product failing to meet quality standards by exceeding colony-forming unit thresholds and potentially through the production of mycotoxins [20,50]. Various factors influence the levels of yeast and mold contamination, which are discussed in the following sections. Testing for the presence of yeasts and molds on cannabis inflorescences prior to harvest is not routinely performed, although research studies have shown that this can provide useful information on the population levels and species that may be present [19]. These populations are influenced by many factors, including the genotype of cannabis being grown, the environmental conditions prior to harvest, particularly temperature and relative humidity, the presence of excessive leaf litter, and the time of year [19].

### 2.8. Flowering Stage IDM Approaches

#### 2.8.1. Cultural and Environmental Management

The increased plant biomass resulting from plant development during the flowering stage creates challenges for the maintenance of consistent environmental conditions, particularly with regard to ambient humidity. Reducing plant densities can significantly lower humidity levels in the greenhouse and also allow for better light penetration and ease of application of disease control products. However, lower plant densities can decrease overall yield per unit area of production [35,51]. A lower ambient relative humidity can also be achieved by increasing air circulation with circulating fans placed near the plants in the weeks leading up to harvest. Maintaining air movement at 0.5–1.0 m/s appears to be an optimal target for microbial suppression in cannabis [51]. Under experimental conditions, enhanced air flow around maturing inflorescences was demonstrated to significantly reduce the populations of various microbes within the tissues of genotype ‘PH’ (Figure 16). This reduction in humidity, combined with appropriate climate control settings, can mitigate the severity of diseases such as bud rot (*B. cinerea*) and powdery mildew during high-risk periods. The cost and practicality of this approach during greenhouse production need to be evaluated, but it provides opportunities for disease management in indoor controlled environments.

In relation to seasonal effects on disease development in the greenhouse, *B. cinerea* bud rot development was shown to be influenced by external vapour pressure deficits that impacted moisture levels in the air and, hence, ambient humidity [35]. To avoid periods of high disease pressure brought on by external environmental conditions, one IDM strategy is to alter the time of seasonal plantings. By scheduling planting and harvest times to avoid periods of high disease pressure brought on by conducive environmental conditions, particularly on desirable but susceptible cannabis genotypes, producers can reduce the impact of seasonal pathogens such as *B. cinerea* [35], as well as reduce the build-up of total inflorescence microbes that are also impacted by seasonal environmental fluctuations (Figure 17).

An alternative approach to reduce disease development is to harvest inflorescences after a shorter crop development period to avoid prolonged exposure to environmental conditions that favour disease development at the maturation stage. For example, harvesting at 6 weeks of inflorescence development instead of 8 weeks can reduce *B. cinerea* bud rot incidence but could result in compromised yield and potency in certain genotypes unless they are close to maturity [35,52]. Areas within a greenhouse that have localized disease or “hot spots” should be identified, followed by the eradication of the affected plants to minimize pathogen spread. The location of the diseased plants should be recorded, and if the causal pathogen is unclear, diagnostic testing should be performed, typically through the submission of samples to a diagnostic laboratory [4,7]. In addition to visualization of these areas with the naked eye, the utility of infrared (IR) and artificial intelligence (AI)-powered scouting technologies could be of value as they have been used in a range of other crops [53,54,55,56,57], but further evaluation of how these technologies could be modified for application to cannabis is needed. A discussion of these technologies is presented in Section 2.10.5 and Section 2.10.6 of this article.

#### 2.8.2. Utility of Disease-Tolerant Genotypes

The cannabis genotype being grown can have a profound impact on the development of certain pathogens, especially under disease-conducive conditions. The impact of genotypes on disease development at the stock cultivation and propagation stages was described previously. A similar significant effect of genotypes on pathogen infection can also be demonstrated at the flowering stage. A comparison of the response of six genotypes to four pathogens is shown in Figure 18. The genotype ‘LO’ showed high susceptibility to powdery mildew but low susceptibility to HLVd, *B. cinerea* bud rot and root pathogens. A second genotype, ‘LB’, showed high tolerance to all four pathogens, while the remaining genotypes varied in their response to these specific diseases. These data were collected from observation trials under natural infection and not from replicated trials. They demonstrate, however, that cannabis producers have the option to select those genotypes that show tolerance to several important pathogens under the specific cultivation conditions of greenhouse production. While the genetic basis for this level of tolerance has not been determined, it indicates there is a basis on which to establish breeding programs that can lead to the development of disease-tolerant cannabis cultivars.

#### 2.8.3. Application of Biological Control Agents

As described for cuttings during propagation and during vegetative growth of plants, biological control agents also show promise in reducing specific diseases at the flowering stage of cannabis plants. The diseases of importance that can be targeted are *B. cinerea* bud rot and powdery mildew. Application of several biological control products and reduced-risk chemicals at weekly intervals as a fine spray, at full label rates, onto developing inflorescences of the genotype ‘PH’ was observed to reduce the development of *B. cinerea* bud rot under both low and high disease pressure resulting from natural infection during the fall growing season (Figure 19).

The most effective product was Rootshield HC^®^ (containing *Trichoderma harzianum*), followed by Regalia^®^ (*Reynoutria sachalinensis*), Double Nickel^®^ (*Bacillus amyloliquefaciens*), Lifegard^®^ (*Bacillus mycoides*), and Prestop^®^ (*Gliocladium catenulatum*). Zerotol^®^ (hydrogen peroxide) did not show an effect (Figure 19). The efficacy of the various biocontrol agents likely stems from their ability to pre-emptively colonize the inflorescence tissues and exert competition against the pathogen, a mode of action also reported on other crops [58,59]. The application of *T. harzianum* was also found to suppress the development of other microbes naturally present within the inflorescences, including *Penicillium* spp., and this was reflected by a reduction in all three categories of microbial counts (Figure 20).

Following this spray trial, a second trail demonstrated that applying *T. harzianum* (Rootshield HC^®^) thrice to the foliage of flowering cannabis plants also reduced the development of powdery mildew compared to untreated plants, as shown in Figure 21. These results indicate that a single biological control agent may target two important diseases affecting cannabis, namely *B. cinerea* bud rot and powdery mildew. *Trichoderma* applications have been shown to suppress powdery mildew in several crops [60,61,62]. The ease of application of the product and the potential to increase microbial counts in inflorescences of treated plants may determine the extent to which cannabis producers are willing to apply biocontrol agents to flowering plants. The ability of other registered biocontrol products to provide a similar disease suppressive activity needs to be assessed.

#### 2.8.4. Application of Reduced-Risk Products

A number of reduced-risk products are available for use on cannabis plants at the flowering stage. During this phase of crop development, care must be taken to avoid damage to inflorescence tissues and to avoid visual quality changes. Products, including Agrotek vaporized sulfur^®^, Regalia Maxx^®^, Suffoil-X^®^, and Milstop^®^, are registered to reduce powdery mildew development [43]. Sulfur is applied via vaporizing pots, a method that ensures uniform dispersal and is commonly used on many other greenhouse crops [31], while the remainder is applied as sprays. In a comparative study to evaluate these and other products for powdery mildew control on flowering cannabis plants, nine products were applied thrice at days 0, 7, and 14 of the flowering period (~60 mL per plant) during the spring season on ‘MP’, a susceptible cannabis genotype, prior to disease appearance. Subsequently, disease severity was rated visually using a leaf infection coverage scale as follows: 0 = 0%, 1 = 1–33%, 2 = 34–66%, 3 = 67–100% (Figure 22a–d). Results showed that Suffoil-X^®^ applied at a rate of 10 mL/L and Regalia Maxx^®^ applied at a rate of 2.5 mL/L were the best preventative products (Figure 22e). In a subsequent trial with the same genotype, flowering plants visibly infected with powdery mildew (disease rating of 1) received one application of seven products at their maximum label rates made at day 42 of the flowering period to evaluate their curative potential. The findings showed that Milstop^®^ applied at a rate of 3 g/L and Cyclone^®^ applied at a rate of 12 mL/L were the best for curative treatments (Figure 22f). The remaining products provided varying levels of disease reduction. No phytotoxicity was observed in any of the treatments. The active ingredients in Milstop^®^ (potassium bicarbonate), Cyclone^®^ (citric and lactic acid), and Suffoil-X^®^ (mineral oil) are all considered to be ‘physical’ in their mode of action, altering leaf surface pH and osmotic pressure or desiccating/coating mycelium and spores [6]. The active ingredient in Regalia Maxx^®^ is an extract from the giant knotweed *Reynoutria sachalinensis* and was shown to be effective against pathogens such as *B. cinerea* and powdery mildew on cannabis as well as on various other crops [63,64,65,66]. This product enhances plant defense responses through the salicylic acid (SA)-dependent pathway by inducing the accumulation of plant defense chemicals such as hydrogen peroxide and the formation of mechanical plant defenses such as callose papillae [66,67]. Additional research is needed to explore the breadth to which Regalia Maxx^®^ can control other pathogens and the duration of the protection offered following application.

A summary of the IDM approaches that can be used against four important pathogens of cannabis is provided in Table 1.

### 2.9. Post-Harvest IDM Approaches

Following the harvest of cannabis inflorescences, they undergo a phase of drying to reduce the moisture content to levels that would minimize the development of microbes [19,20,52], following which they are trimmed and prepared for packaging and stored prior to shipment. During each of the post-harvest processing stages, there is the potential for microbial contamination to be increased, primarily consisting of total yeasts and molds (TYM), total aerobic microbial count (TAMC), and bile-tolerant Gram-negative count (BTGN). Some of these microbes likely originated from the original fresh harvested inflorescences while in the greenhouse or otherwise may have been picked up through contamination during harvesting and post-harvest processing stages. Detailed studies are lacking regarding at which specific stages the levels of microbes may build-up to cause the final product to potentially fail to meet regulatory standards. However, pre-harvest, it has been shown that cannabis genotype and growing conditions can significantly influence TYM build-up; in addition, post-harvest drying methods and handling practices can affect TYM levels [19,20,52]. A number of commonly encountered fungi have been identified on dried cannabis products pre- and post-harvest (Figure 15), and they contribute to TYM levels [19].

The implementation of integrated disease management (IDM) approaches to reduce total yeast and mold (TYM) is complicated by several pre-harvest variables. For example, TYM levels tend to be higher in the summer season than in winter, while certain cannabis genotypes tend to accumulate much higher TYM than others [19]. Post-harvest handling practices also influence TYM levels (hang-dried inflorescences have lower TYM than those that are rack-dried) [19]. Managing these factors to minimize microbial build-up depends on the appropriate application of IDM strategies that were previously outlined for stock plants and during propagation, vegetative growth, and flowering. Post-harvest processing practices, such as reducing moisture by hang-drying plants at a high vapour pressure deficit (VPD) and trimming only after inflorescences are dried, along with thorough cleaning of post-harvest processing equipment using sanitizing agents, can significantly reduce microbial load on inflorescences. Additionally, conducting detailed inspections at each stage of post-harvest processing to detect the presence of molds is critical. This may involve various standard practices, including predefined in-process acceptable quality level (AQL) checks, to ensure that any quality issues are identified and addressed prior to shipment, as is commonly carried out in many food processing plants [68].

Irradiation of cannabis products with gamma and electron beam irradiation has been shown to be an effective option for producers; they can be used to sterilize commercial batches of inflorescences without major changes in quality, but they are costly [69,70,71]. Irradiation is typically used in cases where microbial levels have exceeded regulatory limits or where zero tolerance is recommended, i.e., for medical patients with immunocompromised immune systems that rely on cannabis [20]. Other approaches have been described that require more in-depth studies to demonstrate their commercial utility [72,73]. A summary of the various approaches that can be implemented as a part of an IDM program for greenhouse-cultivated cannabis is presented in Figure 23. These are organized according to the growth stages of the cannabis crop, as described previously. These approaches can be readily implemented, and examples of their successful use have been included in this review. There are additional potential IDM approaches for cannabis that require further research but which have shown potential in other crops, and they are described below.

### 2.10. Future Potential Areas for IDM Development for Cannabis

#### 2.10.1. Evaluation of Endophytes and Microbial Antagonists in Cannabis

Endophytes, consisting primarily of fungal and bacterial species, are present within various tissues and organs of cannabis and hemp plants and vary in species composition, depending on the tissue source, such as roots, stems, petioles, leaves, flowers, and seeds [37,40,74,75,76]. Various plant growth-promoting rhizobacteria, including species of *Bacillus* and *Pseudomonas*, have also been reported to be present in the roots of cannabis plants and can inhibit the growth of root pathogens [40,77]. These endophytes can potentially improve plant growth and development [78,79], although research evaluating their growth benefits in cannabis and hemp plants is currently lacking. Dumigan and Deyholos [37] reported that seed-borne bacterial endophytes, including *Bacillus subtilis* and *B. inaquosorum,* showed inhibitory activity in dual culture assays against fungal pathogens, including *Alternaria* and *Fusarium* species. These endophytes were also present in hemp seeds and included *Bacillus velezensis* and *Paenibacillus polymyxa,* which were also inhibitory to the growth of *Alternaria, Aspergillus, Fusarium,* and *Penicillium* species in vitro. *Pseudomonas* species have also shown growth inhibition of *Fusarium* species in vitro [80]. In previous research, antagonism to *B. cinerea* in dual culture assays was demonstrated for several cannabis-derived endophytes (*Paecilomyces lilacinus* and *Penicillium* spp.) [75] and for several hemp-derived endophytes (*Pseudomonas fulva* and *Pseudomonas orientalis*) [80]. In a study conducted by Gabriele et al. [81] investigating the endophytes present in seeds and young plants of a cannabis cultivar, a unique resistance to the plant’s own antimicrobial compounds was discovered, along with an enhancement of nutraceutical aspects such as polyphenol content and antioxidant activity in the plants. This finding suggests the potential for introducing these endophytes as natural biostimulants and biological control agents against pathogenic microbes, unhindered by the plant’s inherent antimicrobial properties. Such symbiotic relationships underscore the potential of endophytes in cannabis cultivation, but further research is needed to establish their potential applications. The antagonistic properties of endophytic bacteria have been attributed to antibiotic production, host defense response induction, growth promotion, competition, parasitism, and quorum signal interference [82,83,84,85]. Despite these promising studies, however, whole plant assays demonstrating the benefits of these bacteria and other fungal endophytes are presently lacking for cannabis. It should be noted that fungal endophytes can also be present in stem tissues of mother plants, including those shown in Figure 24, and they could negatively impact the health of these plants over time and complicate attempts to initiate tissue cultures using explants from these plants [86]. Inoculation of exposed cut surfaces on stems on cannabis plants with these endophytic fungi showed that species of *Fusarium*, *Penicillium* and *Trichoderma* rapidly colonized the tissues internally and were recovered at distances away from the point of inoculation [17]. Fungal endophytes that are consistently present in stems of cannabis plants include species of *Penicillium* and *Chaetomium*, as well as others (Figure 24). Bacterial endophytes include species of *Bacillus* and *Pseudomonas* [86]. Although commonly recovered from different genotypes of cannabis grown under commercial conditions, the influence of genotype on the frequency of occurrence of these endophytes is unknown. Similarly, the impact of growing conditions, including the substrate used for plant growth, on these internalized microbes has not been determined. Under experimental conditions, the application of a systemic fungicide to growing cannabis plants was shown to reduce the frequency of occurrence of fungal endophytes [86]. This approach was used to reduce the occurrence of endophytic microbes that were encountered as contaminants in tissue culture experiments [86].

Another aspect of potential microbial antagonism against fungal pathogens infecting cannabis that requires research is the diverse microflora that can be present in organic soils compared to conventional hydroponic cultivation. Punja and Scott [87] reported that a diverse range of microbes were recovered from cannabis inflorescences grown in organic soil compared to the cocofibre medium commonly used in hydroponic cannabis production. These communities were comprised of pathogenic, saprophytic, and beneficial microbes. Among the beneficial microbes detected, *Trichoderma harzianum* and *Metharzium anisopliae* are currently used as biological control agents for root disease suppression and insect suppression, respectively. *M. anisopliae* may hold some potential for cannabis pathogen suppression as well [86]. In the context of disease management, similar microbes that originate from organic soils and exhibit general antagonistic properties, such as mycoparasitism, host defense response induction, competition, and antibiotic production, are worthy of evaluation [85,88,89,90]. Cannabis plants grown in ‘living soil’ or growing media amended with ‘compost teas’ may foster greater colonization of roots by these beneficial endophytes, although more research is needed to demonstrate their utility in an IDM program. It is likely that many of these microbes comprise a complex of bacterial species. Caution should be exercised to ensure these microbes do not colonize the inflorescences internally or externally, potentially leading to a failure of the product due to an excessive buildup of microbes.

#### 2.10.2. Tissue Culture Applications for Cannabis

The tissue culture of cannabis has received considerable recent interest in efforts to obtain a source of clean plant materials that can be free of pathogenic microbes, including fungi, viruses, and viroids. Detailed methods have been described from several laboratories, and various techniques have been described [38,86,91,92]. The interest among cannabis producers in utilizing tissue culture methods is to obtain pathogen-free plants and minimize pathogen re-introduction into commercial production facilities. This is particularly relevant in the context of hop latent viroid, which is known to be spread through vegetative cuttings taken from infected mother plants [26]. Meristem tip culture technology has been used for many decades to eliminate the potential for virus introduction in other vegetatively propagated crops, such as potatoes, bananas, and strawberries [93,94,95]. Meristem and shoot-tip culture techniques have been utilized not only for virus elimination but also for rapid clonal multiplication and germplasm preservation of many vegetatively propagated crops [96,97]. In some cases, these methods are augmented with cryotherapy (cold treatment), thermotherapy (heat treatment), chemotherapy (anti-viral chemical treatment), electrotherapy (electrical current treatment), and shoot-tip grafting (micrografting technique) to enhance the chances of obtaining pathogen-free planting materials [98,99,100,101,102]. Research to evaluate the applicability of these methods to obtain pathogen-free planting materials of cannabis, particularly for HLVd, is still in the early stages of evaluation and development. Tissue culture-derived plants can be obtained from meristems and nodal explants of cannabis, resulting in shoot growth of a number of genotypes in vitro (Figure 25). However, confirmation of the eradication of pathogens of importance requires additional research. Hence, while tissue culture approaches hold promise for potential inclusion in an IDM program for cannabis, more effort to generate high frequencies of plants confirmed to be pathogen-free on an economically feasible scale is needed. The confirmation of pathogen-free planting materials could be utilized for certification programs for cannabis, similar to many agriculturally important crops.

#### 2.10.3. Registration of Pathogen Control Products for Cannabis

To evaluate new products aimed at managing fungal pathogens in agricultural crops, screening for pathogen growth inhibition is an important first step. For example, effective concentration (EC_50_) values are determined to establish fungicide levels needed to inhibit 50% of the pathogen’s growth in vitro. However, such studies are less commonly reported for products intended for use on cannabis. EC_50_ studies, which are relatively straight forward to conduct, as demonstrated in Figure 6, are informative about the potential of new products to inhibit the growth of specific pathogens affecting cannabis, especially when followed by whole plant assays. These studies can also determine if there are any secondary effects on biocontrol fungi, such as *Trichoderma* spp. (Figure 6). Recent evaluations of products for powdery mildew control in organic hemp production [103] serve to identify products that may be acceptable for registration in cannabis. Such products could be utilized for pathogen control at the stock plant and propagation stages, which are critical to ensure that the subsequent vegetative and flowering stages do not carry over the pathogen inoculum and to reduce concerns about product residues in the finished flower. Currently, the majority of products registered for use on cannabis can be applied up until harvest, as outlined by Scott et al. [5]. Several commercial products have since been added to the registered list that are not included in [5]. For soil fumigation, Pic Plus Fumigant^®^ (Chloropicrin—85.1%), Chloropicrin 100 Liquid Soil Fumigant^®^ (Chloropicrin—85.1%), and Mustgrow Crop Biofumigant^®^ (Oriental Mustard Seed Meal—100%) can be used pre-plant. For powdery mildew suppression, Vegol Crop Oil^®^ (Canola Oil—96%), Suffoil-X^®^ (Mineral Oil—80%), Purespray FX (Mineral Oil—80%), and General Hydroponics Suffocoat (Canola Oil—96%) can be applied as foliar sprays. For suppression of *B. cinerea*, powdery mildew, and *Sclerotinia sclerotiorum*, Timorex Gold^®^ (Tea Tree Oil—23.8%) can be used. For *Phytophthora* spp. and *Verticillium dahlia* suppression, Foretryx^®^ (*Trichoderma asperellum* strain ICC 012 and *Trichoderma gamsii* strain ICC 080) can be used. While many of these products are different formulations of the same active ingredient, unique products have been added each year since the legalization of cannabis production in Canada. A current list of registered products can be found at Health Canada—Pesticide Label Search (hc-sc.gc.ca).

#### 2.10.4. Nutrient Supplements for Cannabis Disease Suppression

Nutrient amendments have been shown to impact plant susceptibility to infection by a range of pathogens, often reducing disease development through various mechanisms. These involve a wide range of macronutrients and micronutrients. In hydroponic greenhouse cultivation, nutrient levels are carefully monitored to prevent deficiencies; thus, additional nutrient supplements must be approached cautiously to avoid phytotoxicity or imbalances. Formulations and rates are critical factors when considering using these nutrient amendments for disease management [104,105]. The use of nutrient supplements containing copper, silicon, and calcium shows particular promise for cannabis and can be applied via the roots or foliage, as discussed below.

Copper has a long history of use as a bactericide and fungicide for various crops against numerous pathogens since the discovery of the ‘Bordeaux mixture’ in 1885. It disrupts fungal cell membrane integrity and interferes with key enzyme activities, thereby inhibiting pathogen growth and survival [106,107]. Copper can be applied to cannabis plants as root zone drenches, foliar sprays, or seed treatments. For instance, Mayton et al. [108] assessed different seed treatments to manage damping-off caused by *Pythium* and *Fusarium* species on industrial hemp. Seeds treated with a copper-containing product, Ultim^®^ at 0.05 mg Cu/seed, showed efficacy that was comparable to fungicide treatments. Moreover, copper nanoparticles have been successfully applied as dips and foliar treatments on tomatoes and watermelons to reduce *Fusarium* infection [109,110]. A copper formulation, Copper Crop^TM^, reduced powdery mildew on melons [111]. This suppression aligns with the conventional use of copper sulfate pentahydrate as a foliar fungicide on plant species such as roses and dogwood [112,113]. On grapevines, copper citrate effectively reduced *B. cinerea* infections [114]. The diverse range of pathogens suppressed by copper formulations suggests its potential for use in cannabis; however, copper is not currently registered for this purpose.

Silicon is effective against various bacterial, fungal, and viral pathogens, since it can strengthen cell walls via silicon deposits and also induce plant defense responses [115,116]. Scott and Punja [43] reported that multiple weekly sprays of potassium silicate, Silamol^®^, on vegetative cannabis plants significantly reduced powdery mildew development. In contrast, three preventative applications made during the flowering stage to a single genotype, showed no effect in the present study (see Figure 22). Akinrinlola et al. [103] reported that Sil-Matrix^®^, a fungicide with potassium silicate, significantly reduced hemp powdery mildew by 88%. Dixon et al. [117] demonstrated that root-applied silicon at a rate of 600 kg/ha significantly reduced powdery mildew severity in hemp. Similar benefits of silicon supplementation have been observed in crops such as cucumbers, roses, and strawberries [118,119,120,121,122,123]. Further assessments of silicon-containing products for use in cannabis are needed to establish rates and times of application and demonstrate efficacy against pathogens affecting greenhouse crops. 

Calcium application has been shown to reduce pathogen infection by strengthening plant cell walls, thereby providing greater structural integrity against fungal and bacterial infections [124]. However, its effectiveness in reducing pathogens affecting cannabis has not been studied. There are no reports of a direct toxic effect of calcium-containing compounds on fungal pathogens affecting cannabis, suggesting that its action may stem from reducing host susceptibility or through other mechanisms. In some crops, root-zone supplementation of calcium nitrate was reported to reduce *B. cinerea* severity in beans and tomatoes, although higher doses increased disease in beans [125]. Supplementing roses with calcium nitrate and adding calcium chloride or calcium sulfate to solutions for harvested flowers reduced *B. cinerea* incidence under conducive disease conditions [126]. Similarly, increasing calcium and reducing nitrogen levels in the irrigation water for sweet basil plants reduced both sporulation and infection severity of *B. cinerea* [127]. Whether enhanced calcium supplementation can influence the development of *B. cinerea* in cannabis plants remains to be determined. In addition, its potential for reducing infection by root pathogens such as *Fusarium* and *Pythium* should be explored.

#### 2.10.5. Artificial Intelligence (AI) Technologies for Cannabis Disease Detection

The use of recently developed robotic and imaging technologies for scouting for disease presence has garnered interest from cannabis producers. Various options are available with pros and cons, depending on the greenhouse scale, layout, and operations. For small-scale greenhouses, fixed crop monitoring cameras or AI-powered phone scouting apps are often utilized. Several cannabis-focused scouting apps include Koppert’s Natutec Scout app (https://www.koppert.com/natutec-scout/) (accessed on 22 January 2024), BioBest’s Crop-scanner^TM^ app (https://www.crop-scanner.com/) (accessed on 22 January 2024), GrowDoc AI’s app (https://growdoc.ai/) (accessed on 22 January 2024), and the IPM Scoutek^TM^ app (https://ipmscoutek.com/) (accessed on 22 January 2024). In contrast, larger greenhouse operations, with a more consistent layout, have trialled autonomous robotic scouting carts or booms with cameras attached to crop carts, such as IUNU’s LUNA AI scouting above-crop cameras or scouting carts (https://iunu.com/luna-ai) (accessed on 22 January 2024), Ecoation’s OKO or ROYA scouting carts (https://www.ecoation.com/integrated-pest-management) (accessed on 22 January 2024), and Budscout AI’s Budscout above-crop cameras (https://budscout.ai/budscout/) (accessed on 22 January 2024). The challenge for AI and imaging solutions currently is that they may not reliably distinguish symptoms caused by various pathogens from nutrient deficiencies and other environmental stressors. Supplemental and tailored training are likely needed to achieve accurate results. In the broader agricultural sector, significant progress is being made in the robot AI-assisted vision space [56]. An example of the training process and customizability of AI scouting technology is demonstrated in the following two studies. Anagnostis et al. [55] aimed to build a fast and accurate object detection system to identify anthracnose-infected leaves (by *Colletotrichum* spp.) in a commercial walnut orchard. The study involved segmenting high-resolution images into smaller sub-images and training an object detector to recognize disease-specific features. The deep learning approach achieved high accuracy under real-field conditions. Similarly, Mahmud et al. [54] focused on developing an innovative machine vision system to accurately detect powdery mildew in strawberry fields. This system utilized real-time image processing and artificial neural networks (ANNs) to distinguish diseased leaves from healthy ones. The study demonstrated the system’s adaptability to field conditions and showed high accuracy in detecting powdery mildew. These examples point to the potential for applications in integrated disease management and early disease intervention in different agricultural settings.

#### 2.10.6. Infrared (IR) Technologies for Cannabis Disease Detection

Infrared imaging (IR), a spectrum used in some remote sensing technologies, identifies variations in crop or leaf temperatures to reflect reduced transpirational activity or metabolic functions, signaling the potential presence of stressors, including disease [53,57,128]. In cannabis, detection of leaf surface temperature changes due to root diseases or poor root development when tested at different developmental stages can be captured with handheld devices such as a FLIR E8 Pro^TM^ infrared camera (https://www.flir.ca/products/e8-pro/?vertical=condition+monitoring&segment=solutions) (accessed on 10 January 2024). This method showed definitively that poorly developed root systems on affected plants were correlated directly with a reduced rate of transpiration and, hence, a build-up of leaf surface temperature that was detectable with the IR camera (Figure 26).

However, when powdery mildew-infected plants or cannabis plants affected by hop latent viroid were similarly compared to healthy plants using an IR camera, these plants did not show a corresponding reduced transpirational activity pattern, suggesting that the IR camera was unable to detect physiological changes in these diseased plants. It is unknown whether infrared or other spectrums could be used to effectively detect hop latent viroid; limited research has been carried out on virus detection with infrared, but there may be potential applications [129,130]. Vagelas et al. [131] utilized a low-cost infrared camera and a standard RGB web camera to analyze vine, chrysanthemum and rose leaves that had been infected with various fungi. Results showed that infected leaves exhibited temperature deviations from uninfected ones, which occurred before visible symptoms developed. Specifically, infected vine and rose leaves showed a decrease in temperature, while chrysanthemum and another set of rose leaves demonstrated an increase, compared to healthy tissue. Lindenthal et al. [132] used infrared thermography to detect downy mildew infection in cucumbers. In controlled environments, the study showed that the maximum temperature difference in a leaf could be used to distinguish between healthy and infected tissues. Under natural environments, while leaf temperatures and transpiration rates were similar in both healthy and infected plants, diseased leaves showed more varied transpiration rates depending on the severity of the symptoms. Liaghat et al. [133] utilized Fourier transform infrared (FT-IR) spectroscopy to detect *Ganoderma* infections in oil palm trees. This method involved analyzing leaf samples from both healthy and infected trees, examining the infrared spectra of these samples, and using a statistical model for classification. The researchers successfully identified differences linked to the disease, accurately identifying infected trees at early, symptomless stages. Therefore, there is a growing body of evidence to demonstrate that IR approaches could be applied to cannabis for early detection of infection by foliar pathogens, but additional studies are required to validate this approach.

#### 2.10.7. Electronic Nose Technologies for Cannabis Disease Detection

The use of “electronic nose systems” (e-nose systems), also known as electronic olfactory systems, for the early detection and diagnosis of plant diseases across various crops is receiving increased attention. The technology involves multiple sensors sensitive to a variety of volatile compounds that generate electrical signals upon exposure to these molecules, which are then digitized and analyzed using machine learning-based pattern recognition algorithms. By leveraging the unique volatile organic compound (VOC) signatures emitted by plants under disease stress, the technology compares detected patterns to known odour profiles to identify the presence and intensity of diseases in an air sample. This approach has demonstrated potential for the early detection and diagnosis of plant diseases across various crops, providing a rapid, non-invasive, and field-deployable solution [53,134,135]. However, despite its advantages in non-destructive and bulk sampling, e-nose technology is considered less sensitive and specific than traditional diagnostics like PCR, suggesting its role is currently best suited as a supplementary tool in an IDM diagnostics program rather than a stand-alone solution [136]. Numerous proof-of-concept studies have applied different methodologies to detect specific pathogen-induced VOC signatures. The Bloodhound^®^ ST214′s efficacy in detecting disease presence by analyzing VOCs emitted by tomato plants infected with powdery mildew (*Oidium neolycopersici*) in greenhouse settings compared to healthy controls was demonstrated [137]. Similarly, a low-cost, portable e-nose combined with machine learning algorithms was used to accurately detect *Fusarium oxysporum* in tomato plants and soil samples [138], while Sun et al. [139] were successful in detecting *B. cinerea* infection in tomatoes. A PEN 3, Win Muster Air-sense Analytics E-nose was used to identify infections caused by three fungi—*Botrytis* spp., *Penicillium* spp., and *Rhizopus* spp.—in strawberries [140]. A custom-built e-nose device was used for early detection of fungal infections in garlic, distinguishing between garlic infected by different fungi (*Fusarium oxysporum*, *Alternaria embellisia*, and *Botrytis allii*) [141]. Hazarika et al. [142] utilized the Alpha MOSFOX 3000 e-nose system to identify Khasi Mandarin orange plants infected by citrus tristeza virus (CTV) with high accuracy by analyzing essential oils extracted from leaves. These studies collectively highlight the versatility of e-nose systems in agricultural applications, underscoring their potential utility in diagnosing cannabis diseases. However, to date, there have been no studies to demonstrate whether pathogen-induced volatile compounds can be separated from the natural constituents present in healthy cannabis plants and whether this technology can be used for early disease prediction.

#### 2.10.8. Induction of Plant Defense Responses in Cannabis

The potential for inducing plant defense responses before pathogen infection has not yet been developed to a practical level for cannabis. However, as discussed earlier, Regalia Maxx (*Reynoutria sachalinensis*), when applied to cannabis plants, can reduce pathogens such as *B. cinerea* and powdery mildew, confirming reports in the literature of the efficacy through presumed induction of defense responses and pathogen reduction [63,64,65,66,67]. Weekly applications are recommended to ensure the ongoing protection of cannabis plants. The role of endophytic microbes in the growing medium in promoting plant health and reducing pathogen infection in cannabis awaits more research. Previous reports demonstrated the defense-boosting properties of endophytic organisms on various plant species [82,83,84,85,143]. The utility of a biological control product containing *Trichoderma* spp. appears to be promising when applied preventatively to the root zone or to inflorescences; however, additional research is necessary to establish whether the induction of defense responses in cannabis can be confirmed as it has been in previous studies on numerous plants [144,145]. The application of compounds such as salicylic acid and jasmonic acid to cannabis plants to potentially induce disease resistance is worthy of study.

## 3. Conclusions

This comprehensive review of integrated disease management (IDM) approaches for greenhouse-cultivated cannabis underscores the significance of developing a multifaceted approach to control the various pathogens of economic concern. The review highlights the importance of pre-emptive measures, including a selection of disease-tolerant genotypes and the use of stringent sanitation practices, in minimizing pathogen incidence. The utilization of biological control agents and reduced-risk products, and the modification of cultural and environmental conditions, have shown promising results in suppressing *B. cinerea* bud rot, powdery mildew, *Pythium* and *Fusarium* root diseases, and hop latent viroid-causing stunt disease. Moreover, the exploration into alternative strategies, including the utility of endophytes, tissue culture, nutrient supplementation, and technology-aided scouting, offers potentially new avenues for enhancing plant health. This review underscores the need for studies on plant defense response induction and modes of action of biological control agents. It shows the dynamic nature of IDM in cannabis cultivation and emphasizes the continuing need for research and adoption of sustainable strategies to meet the evolving challenges in disease management within the greenhouse cannabis industry. Such strategies should receive support from governmental regulatory agencies to ensure they meet the criteria set forth by the appropriate jurisdictions. The flexibility in allowing additional disease management products to be registered for use by cannabis producers is essential to allow the industry to meet the continual challenges imposed by plant pathogens.

## Figures and Tables

**Figure 1 plants-13-00786-f001:**
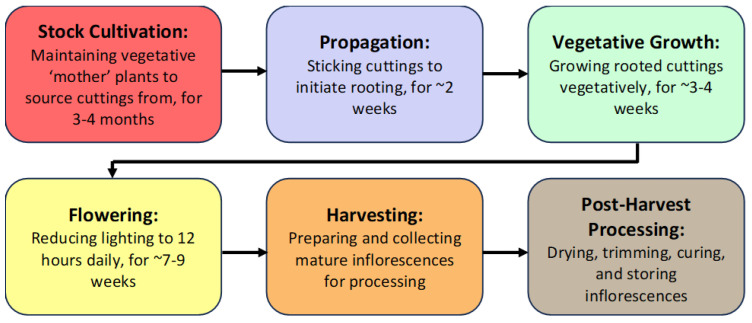
The different stages of cannabis production under greenhouse conditions. Each crop cultivation cycle from propagation to harvest spans ~12–15 weeks. This is followed by a final stage of post-harvest processing that includes drying, trimming, curing, and storage.

**Figure 2 plants-13-00786-f002:**
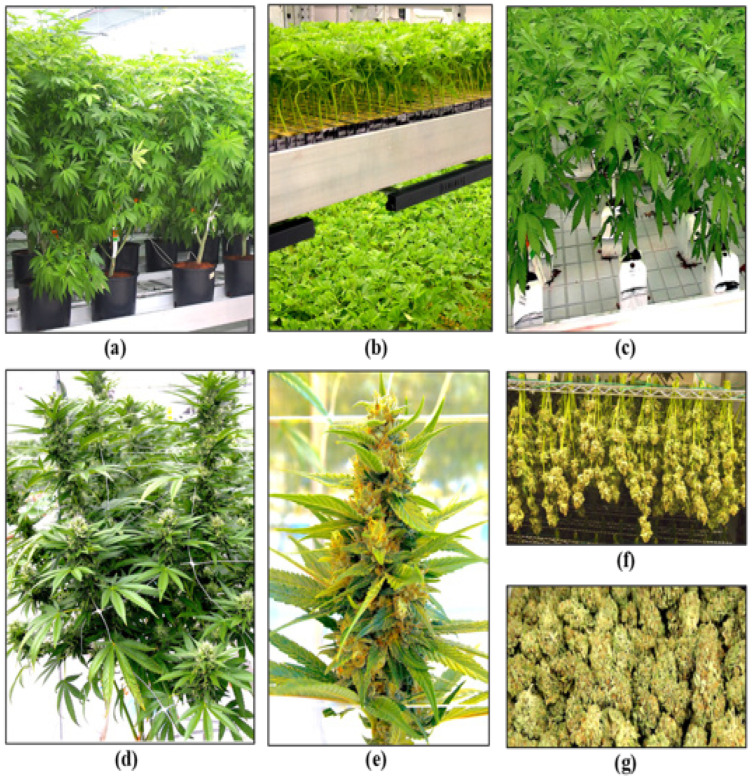
The stages of cannabis crop development. (**a**) Stock plants. (**b**) Rooting of cuttings. (**c**) Vegetative plants. (**d**,**e**) Flowering plants. (**f**) Harvested inflorescences being hung to dry. (**g**) Bucked and trimmed inflorescences.

**Figure 3 plants-13-00786-f003:**
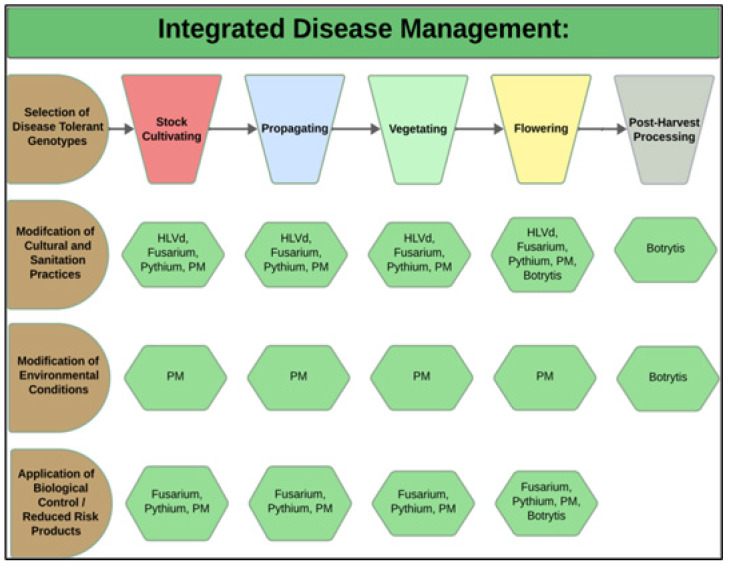
Integrated disease management strategies (left panel, in brown) are developed according to the crop development stage (top panel). The hexagons (in green) illustrate the specific diseases being targeted, which are discussed in more detail below. HLVd = hop latent viroid; PM = powdery mildew; *Botrytis cinerea* = bud rot.

**Figure 4 plants-13-00786-f004:**
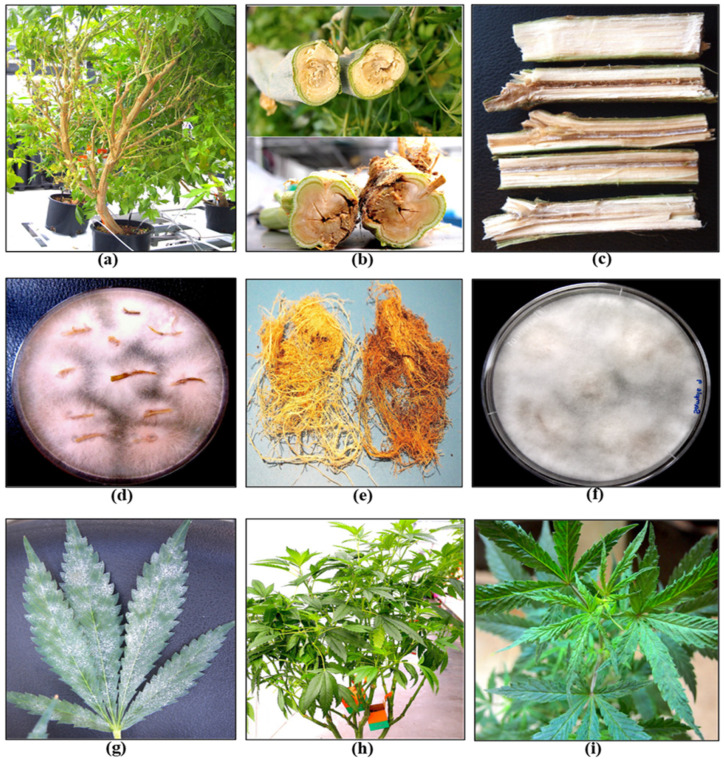
Symptoms of infection by a range of pathogens commonly observed in cannabis stock plants. (**a**) Declining growth with reduced vigor in a 7-month-old plant. (**b**,**c**) Internal stem discolouration due to *F. oxysporum* infection. (**d**) Isolation of colonies of *F. oxysporum* from diseased tissues. (**e**) Browning of roots due to *Pythium* infection. (**f**) Isolation of *Pythium* colonies from diseased roots. (**g**) Powdery mildew infection on the upper surface of leaves. (**h**,**i**) Infection by hop latent viroid causing reduced vigor and curling of young leaves.

**Figure 5 plants-13-00786-f005:**
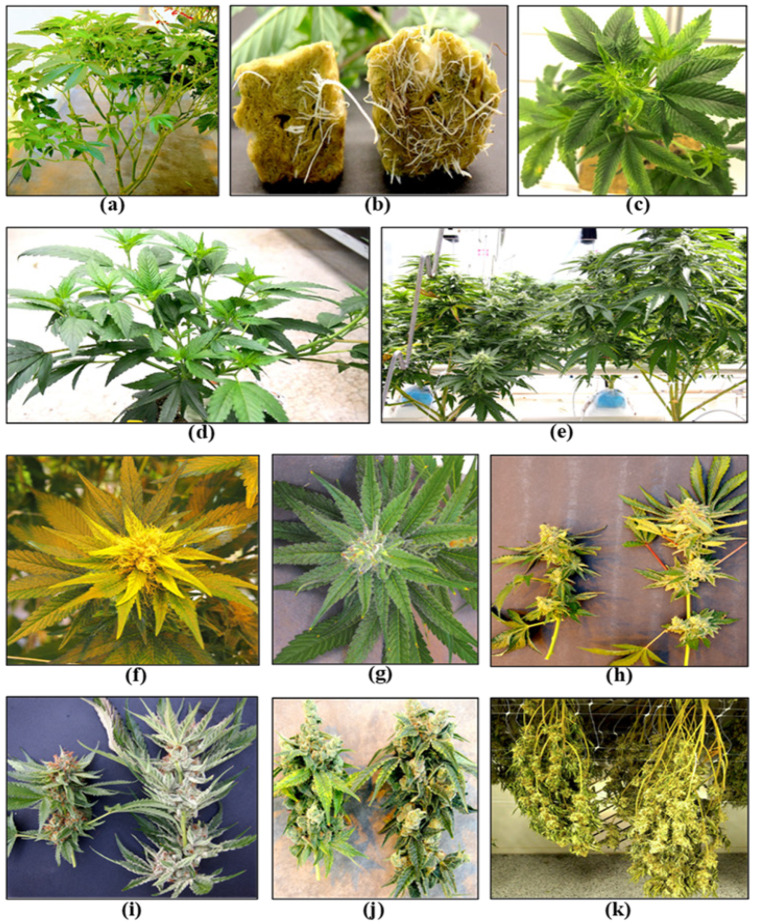
Symptoms of hop latent viroid infection during propagation, vegetative growth and flowering stages of the cannabis crop cycle. (**a**) Infected stock plants may show unthrifty growth and smaller leaves. (**b**) Comparison of root development on cuttings derived from an HLVd-infected stock plant (left) and a healthy plant (right). (**c**) Vegetative plants may show curling and distortion of the youngest leaves. (**d**) Lateral branching may be seen on HLVd-infected vegetative plants. (**e**) Stunted growth of HLVd-infected flowering plant (left) compared to a healthy plant (right). (**f**,**g**) HLVd-infected inflorescence with yellowing compared to a healthy one, respectively. (**h**–**j**) Reduced inflorescence development in three different genotypes of cannabis resulting from HLVd infection. (**k**) Dried inflorescences from an HLVd-infected plant (left) compared to a healthy plant (right). In all comparison photos, the infected plant is shown on the left.

**Figure 6 plants-13-00786-f006:**
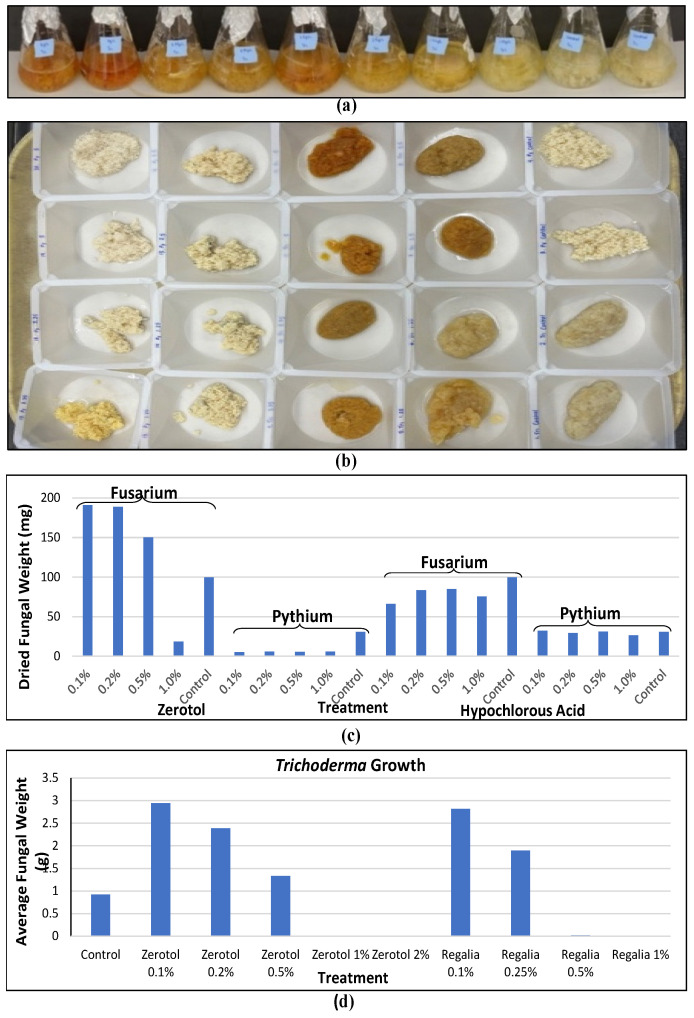
The effect of reduced-risk products on pathogen growth can be evaluated under laboratory conditions by testing a range of concentrations in liquid culture medium. (**a**) Example of fungal growth in potato dextrose broth containing a range of concentrations of individual products. (**b**) Growth is measured by obtaining mycelium dry weights after a 7-day exposure. (**c**) The effect of Zerotol^®^ and hypochlorous acid (1000 ppm) on growth of two pathogens at increasing concentrations from 0.1% to 1.0%. Both *Fusarium* and *Pythium* growth is reduced at higher concentrations, but growth of *Pythium* shows greater sensitivity compared to *Fusarium*. (**d**) Growth of *Trichoderma* can also be reduced by the presence of specific compounds when added to the culture medium.

**Figure 7 plants-13-00786-f007:**
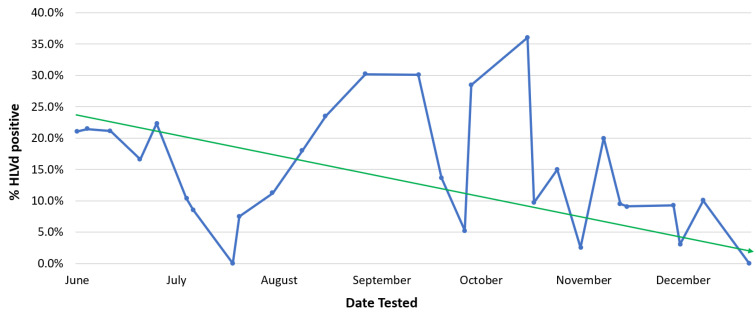
The impact of eradication of HLVd-infected stock plants on the frequency of positively infected plants over a 6-month duration. The blue line shows the actual incidence of infected plants, which fluctuates over time. The solid green line is the general trend that shows a decline in number of infected plants. Presence of the viroid in infected plants was confirmed by RT-PCR. The data shown are from the 2022 growing season.

**Figure 8 plants-13-00786-f008:**
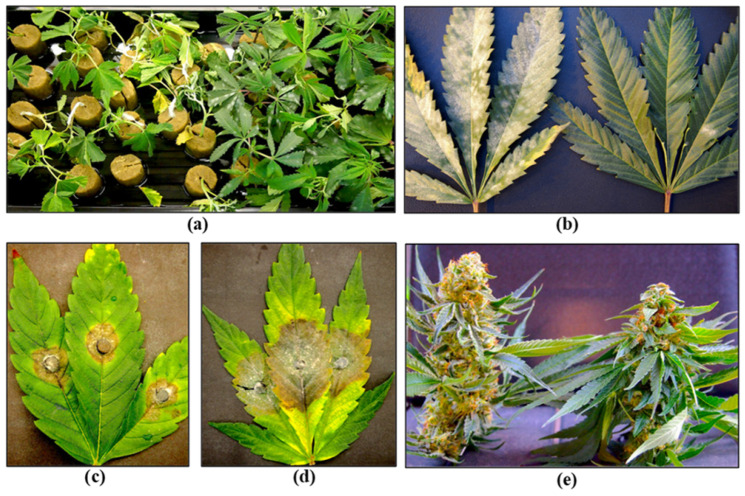
Examples of cannabis genotypes that exhibit a level of disease tolerance to different pathogens. (**a**) *Fusarium* damping-off, with susceptible genotype on the left and tolerant genotype on the right. (**b**) Powdery mildew, with susceptible genotype on the left and tolerant one on the right. (**c**,**d**) *Alternaria* leaf blight, with tolerant genotype on the left and susceptible one on the right. (**e**) *B. cinerea* bud rot, with tolerant genotype on the left and susceptible one on the right.

**Figure 9 plants-13-00786-f009:**
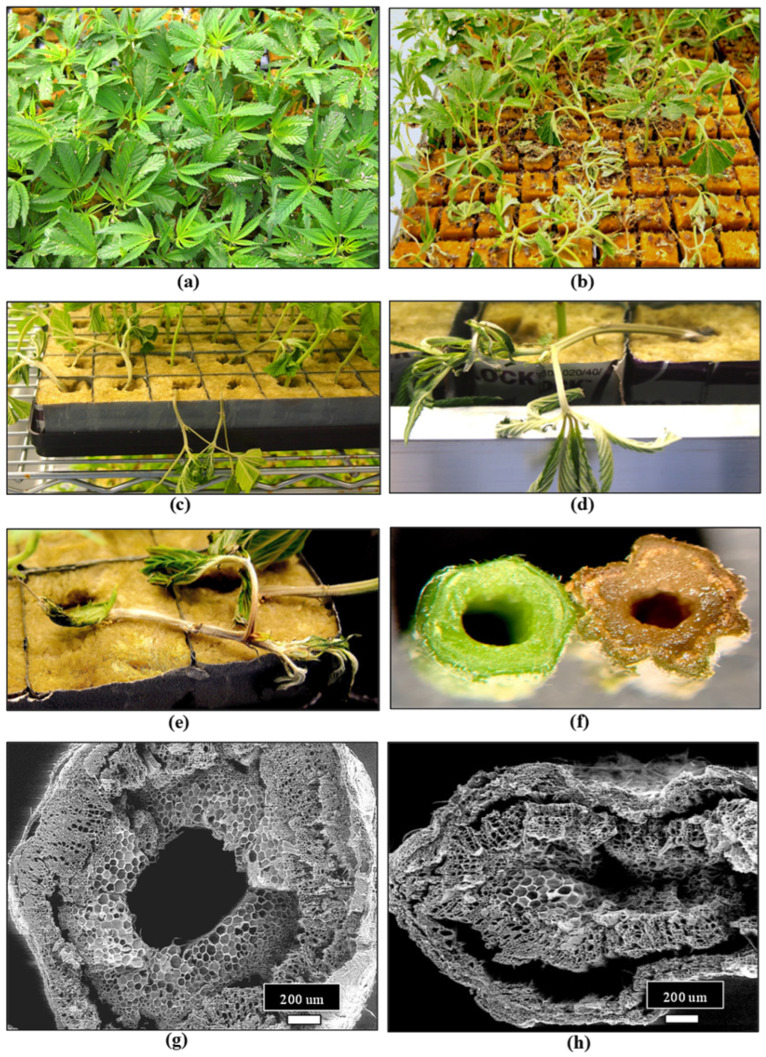
Propagation of cannabis from vegetative cuttings and development of *Fusarium* damping-off. (**a**) A tray of healthy cuttings. (**b**) A tray of cuttings infected with *Fusarium oxysporum*. (**c**–**e**) Close-up views of damped-off cuttings. (**f**) A cross-sectional view of the stem of a healthy cutting (left) compared to a diseased one (right) in which tissue browning can be seen. (**g**) A scanning electron microscopic view of a section through the stem of a healthy cutting. The central pith can be seen. (**h**) A collapsed stem of a diseased cutting viewed through the scanning electron microscope. The central pith has collapsed, as well as surrounding cells.

**Figure 10 plants-13-00786-f010:**
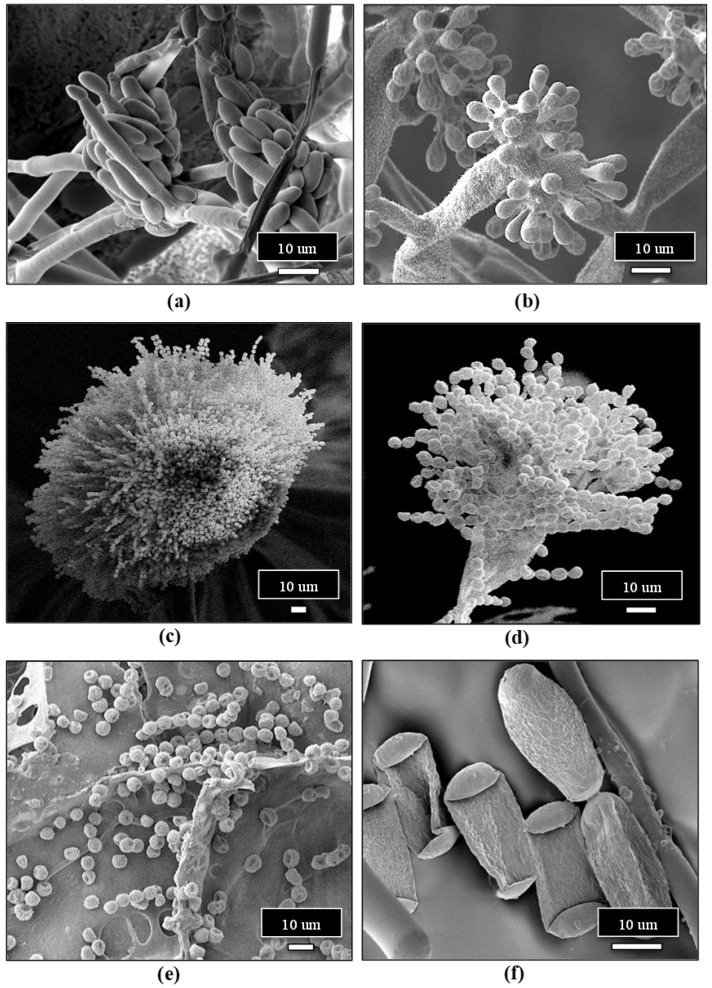
Spores of a range of pathogens that can affect cannabis plants at various stages of crop growth. (**a**) *Fusarium oxysporum* micro-conidia. (**b**) *B. cinerea* spores developing on conidiophores. (**c**) Large cluster of spores of *Aspergillus* spp. (**d**,**e**) Chains of spores of *Penicillium* spp. developing on a conidiophore. (**f**) *Golovinomyces ambrosiae* spores. Scale bar = 5 µm in all photos.

**Figure 11 plants-13-00786-f011:**
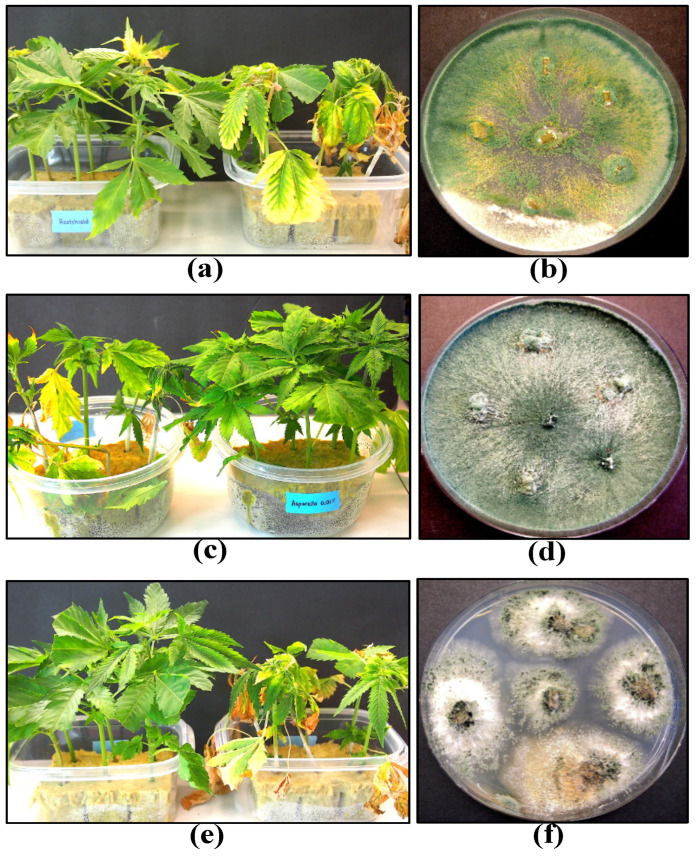
Application of biological control agents provides protection to cannabis cuttings against *Fusarium* damping-off. (**a**) Rootshield-treated cuttings (left) show greater survival compared to pathogen-only cuttings (right). (**b**) Growth of *Trichoderma harzianum* from Rootshield-treated cuttings. (**c**) Asperello-treated cuttings (right) show greater survival compared to pathogen-only cuttings (left). (**d**) Growth of *Trichoderma asperellum* from Asperello-treated cuttings. (**e**) Prestop-treated cuttings (left) show greater survival compared to pathogen-only cuttings (right). (**f**) Growth of *Gliocladium catenulatum* from Prestop-treated cuttings. Recovery of all biological control agents was made on potato dextrose agar medium as shown.

**Figure 12 plants-13-00786-f012:**
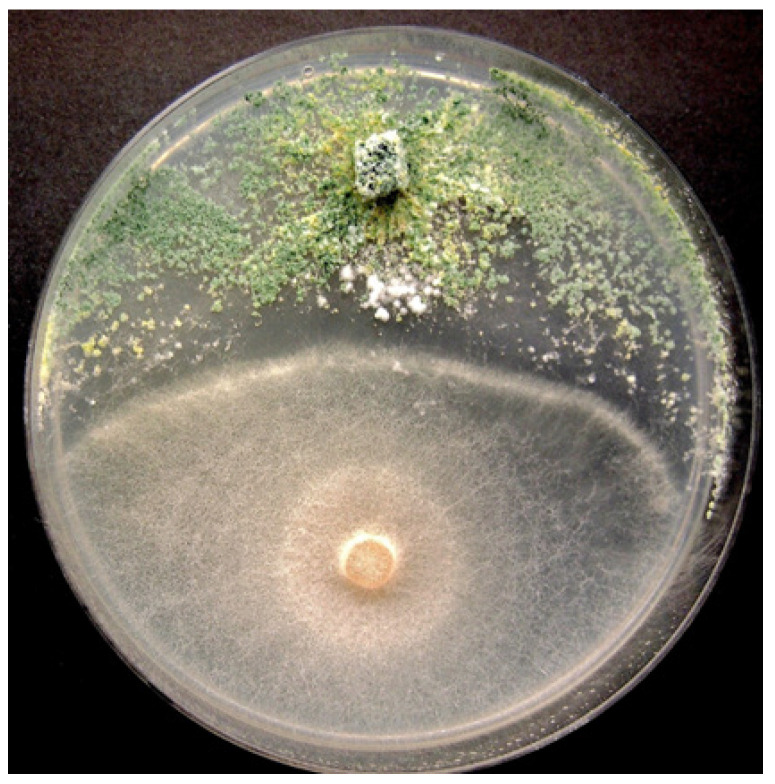
Growth of *T. asperellum* (**top**) is observed to stop the growth of *Fusarium oxysporum* (**bottom**) when both are placed on a Petri dish containing potato dextrose agar medium. After a few days, the biocontrol agent continues to grow and inhibits further growth of the pathogen, indicating its suppressive activity.

**Figure 13 plants-13-00786-f013:**
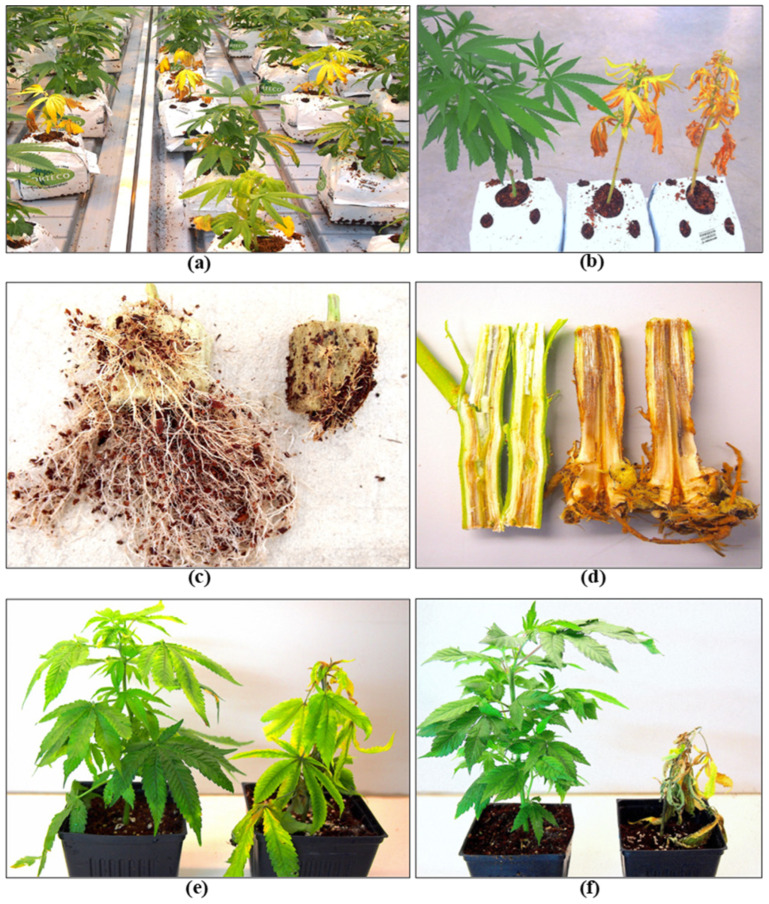
*Pythium* and *Fusarium* infection in vegetative plants of cannabis. (**a**) Symptoms of yellowing of the foliage are indicative of root infection by these pathogens. (**b**) Death of rooted cuttings due to *Fusarium* infection. (**c**) Root development on healthy plant (left) compared to one infected by *Fusarium* (right). (**d**) Internal stem discolouration is indicative of infection by *Fusarium*. (**e**,**f**) Infection by *Pythium* can cause significant stunting of plant growth and death (right) compared to healthy plants (left).

**Figure 14 plants-13-00786-f014:**
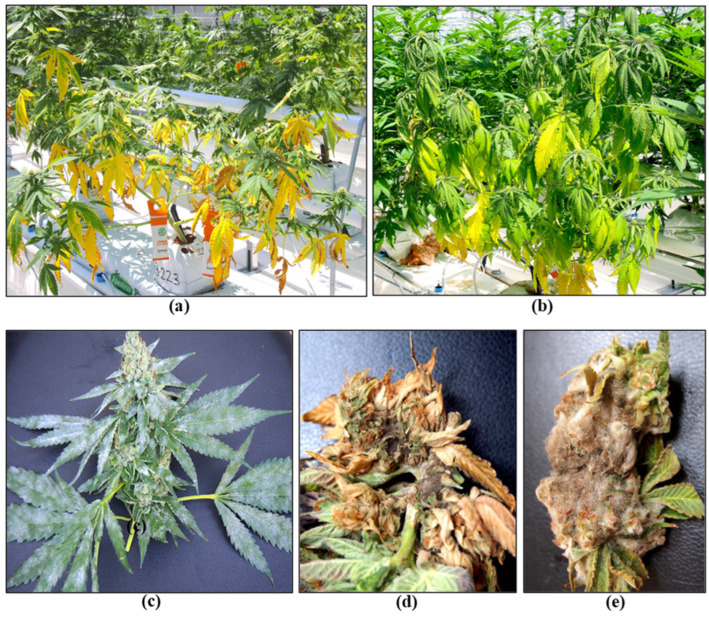
Symptoms due to pathogen infection in flowering cannabis plants. (**a**) Yellowing of the foliage and stunted growth due to infection by *Fusarium*. (**b**) Wilting of plants and yellowing of foliage due to infection by *Pythium*. (**c**) Powdery mildew development on inflorescences and surrounding leaves. (**d**,**e**) Bud rot caused by *B. cinerea* destroys the inflorescence.

**Figure 15 plants-13-00786-f015:**
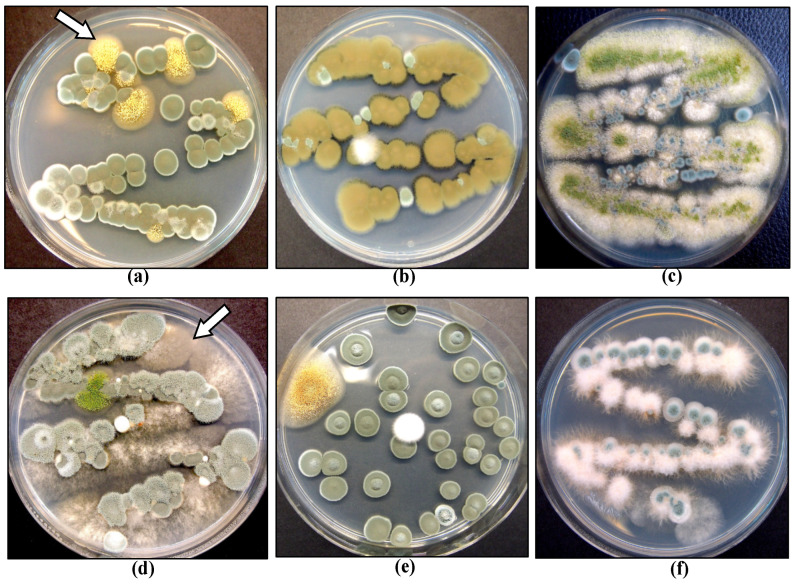
The most commonly recovered fungi from inflorescences of cannabis plants prior to harvest. The Petri dishes show the results from the swabbing of samples and plating onto an agar medium that allows growth of yeasts and molds to occur. (**a**) Green colonies of *Penicillium* with yellow colonies of *Aspergillus* (arrow). (**b**) Brown colonies of *Cladosporium.* (**c**) Mixture of *Aspergillus* (green) with small blue colonies of *Penicillium*. (**d**) Green colonies of *Penicillium* with gray growth of *B. cinerea* (arrow). (**e**) Colonies of *Penicillium*. (**f**) Pink colonies of *Fusarium* with blue colonies of *Penicillium.* Photos were taken after 7 days.

**Figure 16 plants-13-00786-f016:**
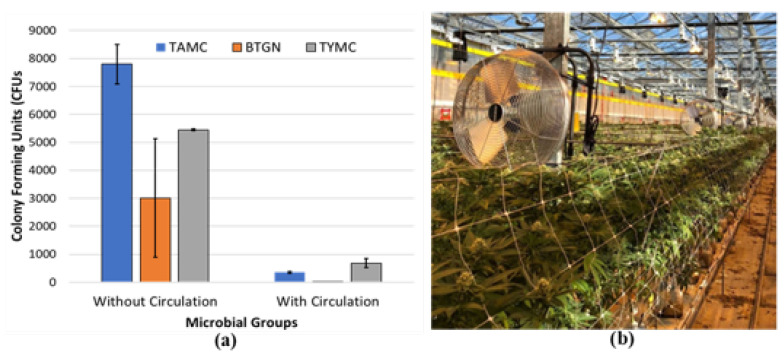
(**a**) Effect of enhanced air flow around cannabis plants using circulating fans on total colony-forming units of microbes in these tissues. Vertical bars show total colony-forming units of total aerobic count (TAMC), bile-tolerant Gram-negative count (BTGN), and total yeast and mold count (TYMC) with and without air circulation. (**b**) Fans were positioned 35 cm above the crop canopy to circulate air continuously at ~7 m/s over ~40 plants, beginning in week 2 of the flowering period until harvest. The trial was replicated three times in different greenhouse compartments. Inflorescences were dried prior to microbial analysis.

**Figure 17 plants-13-00786-f017:**
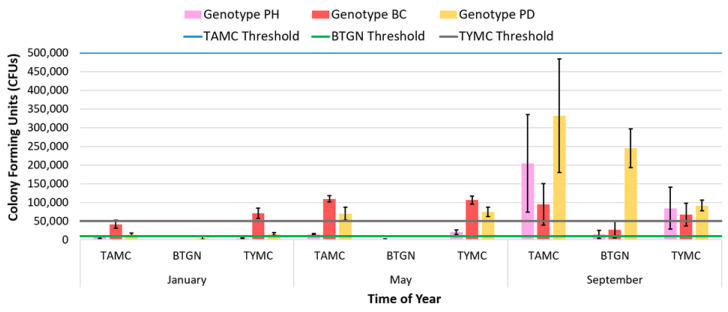
Influence of cannabis genotype and time of year (season) on total microbes present in dried cannabis inflorescences. Vertical bars denote total aerobic microbial count (TAMC), bile-tolerant Gram-negative count (BTGN) and total yeast and mold count (TYMC). Samples were taken from three genotypes during three harvests in each season (fall, winter, and summer seasons) of the same year. Highest microbial counts were observed in the September harvest period, corresponding to late-summer production. The failure thresholds for each microbial group are shown by the horizontal lines. Genotype ‘PD’ contained the highest microbial levels, demonstrating the importance of genotype x environment interactions.

**Figure 18 plants-13-00786-f018:**
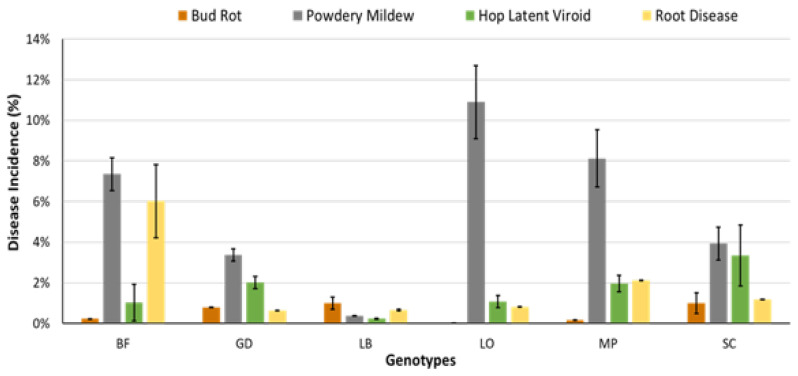
Comparison of disease incidence on six cannabis genotypes to four pathogens, demonstrating variation in susceptibility to *B. cinerea* bud rot, powdery mildew, hop latent viroid and *Pythium* or *Fusarium* root diseases. Incidence data were obtained from scouting of crops conducted during the cultivation of batches of genotypes in comparable greenhouse compartments over three production cycles in the summer season. Disease incidence was assessed based on visual symptoms. Genotype LB shows low susceptibility to all pathogens, while genotypes LO and MP are highly susceptible to powdery mildew.

**Figure 19 plants-13-00786-f019:**
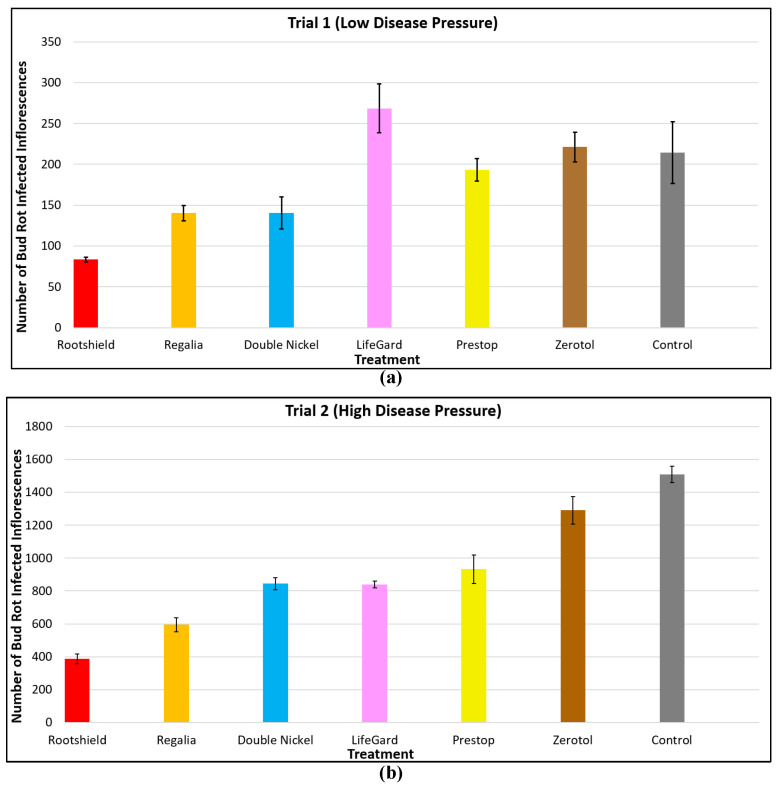
Comparative efficacy of six biological control products and reduced-risk chemicals on *B. cinerea* bud rot development on flowering cannabis plants. Three applications were made at weeks 2, 3, and 4 of the flowering period at maximum label rates. The sprays were applied to 216 plants using a robotic pipe rail sprayer that delivered ~60 mL of product to each plant. Disease assessments were made at harvest (week 8) in a greenhouse compartment with low and high *B. cinerea* bud rot pressure from natural inoculum. (**a**) Low disease pressure flower room; (**b**) high disease pressure flower room. Error bars show standard error of the mean.

**Figure 20 plants-13-00786-f020:**
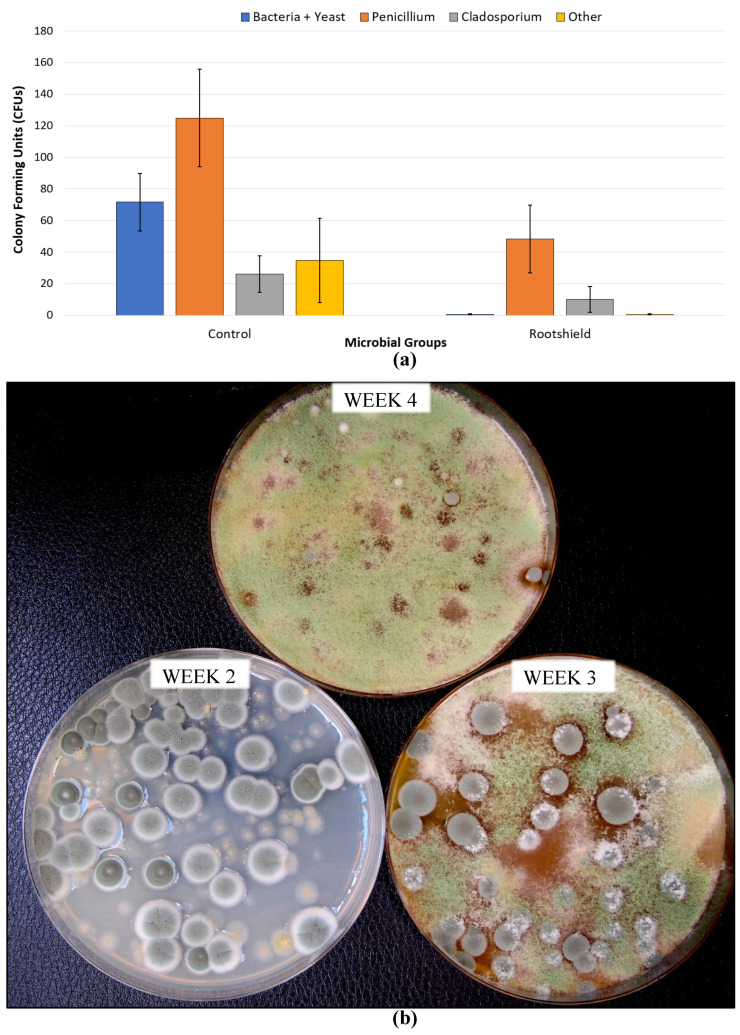
Effect of Rootshield HC^®^ (*T. harzianum*) applications made at weeks 2, 3, and 4 of the flowering period on final microbial levels in harvested cannabis inflorescences. (**a**) Total counts of all microbes in both untreated and sprayed plants are shown. Total microbes were reduced following Rootshield applications. (**b**) The growth of microbial colonies after blending the treated inflorescences in distilled water and subsequent plating onto agar medium. A comparison is shown of samples following applications of Rootshield made at weeks 2, 3, and 4 of the flowering period. Samples treated at week 4 show maximum suppression of *Penicillium* growth compared to week 2, where there was no suppression and no colonies of *Trichoderma* were recovered.

**Figure 21 plants-13-00786-f021:**
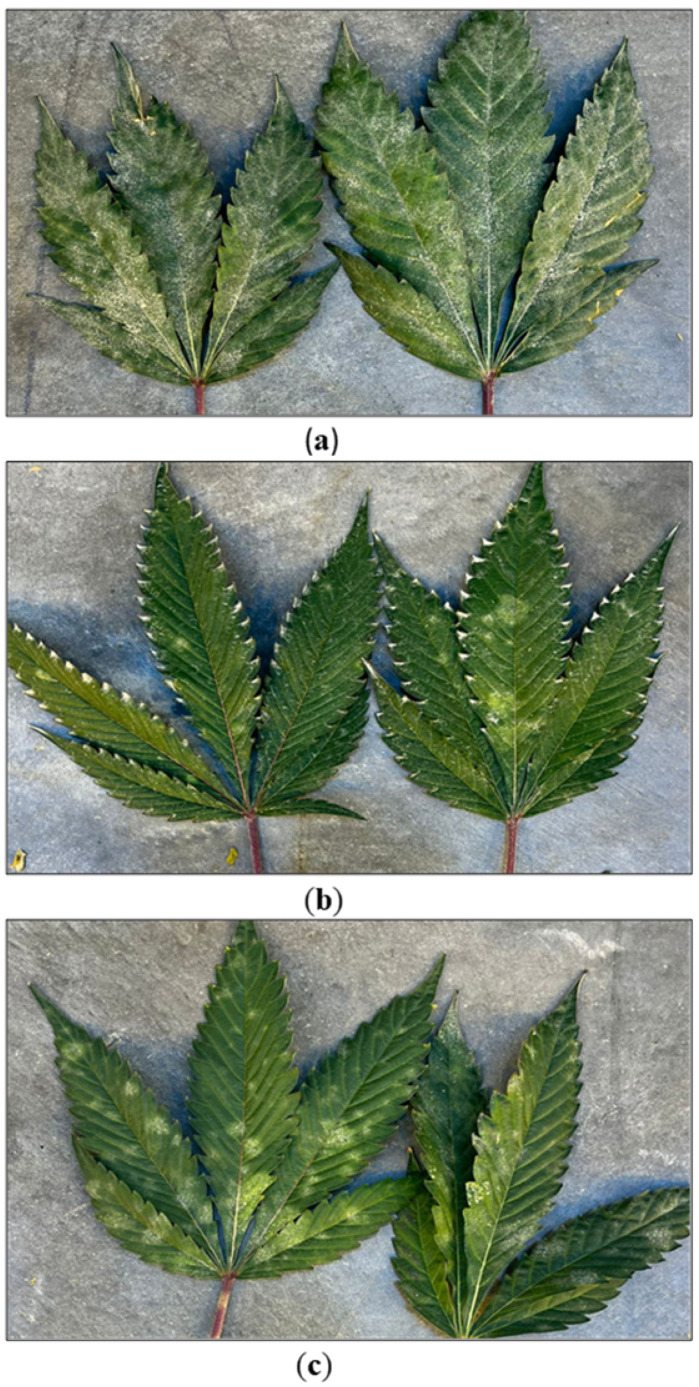
Effect of Rootshield HC^®^ applications on development of powdery mildew on cannabis leaves. Three weekly applications were made to the foliage of flowering plants as preventative treatments and compared to an untreated control and a water control. (**a**) Untreated control leaves. (**b**) Rootshield HC^®^-treated leaves. (**c**) Water-treated leaves. Rootshield applications visibly reduced the development of powdery mildew colonies.

**Figure 22 plants-13-00786-f022:**
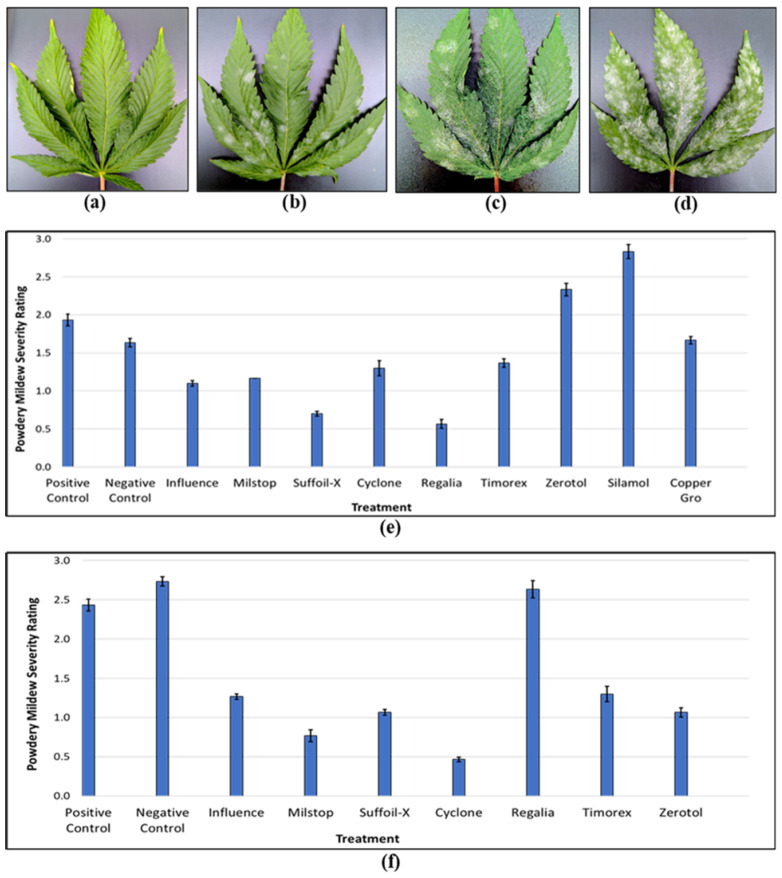
Comparative efficacy of reduced-risk products at managing powdery mildew development on cannabis genotype ‘MP’. (**a**–**d**) Disease was rated according to the scale shown, from 0 (**a**) to 3 (**d**). (**e**) Products were applied as preventative treatments at days 0, 7, and 14 of the flowering period. (**f**) Products were applied as a curative treatment once at day 42 of the flowering period after the onset of disease development. The trials were conducted during the spring growing season.

**Figure 23 plants-13-00786-f023:**
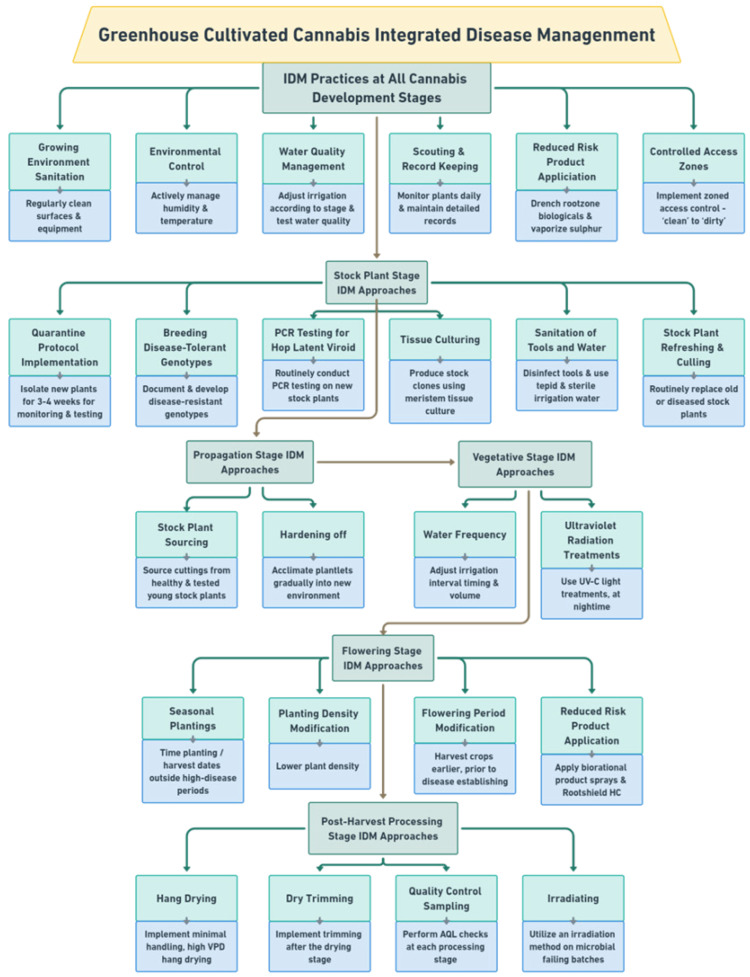
Operational flow chart for various IDM approaches that can be incorporated into an IDM program according to cannabis cultivation stage.

**Figure 24 plants-13-00786-f024:**
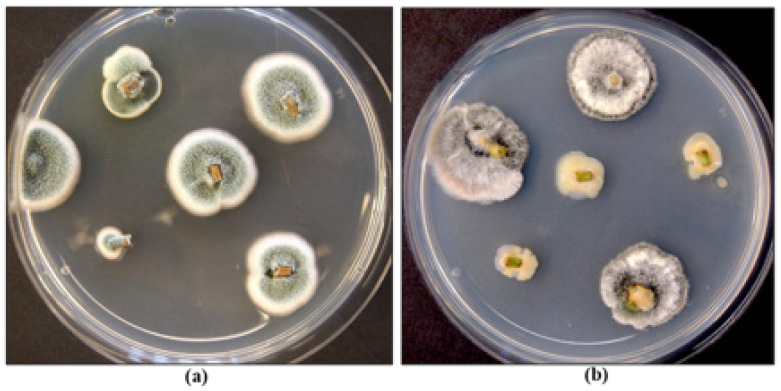
Examples of endophytic fungal and bacterial species recovered from cannabis stem segments following sterilization. (**a**) Petri dish with *Penicillium* species; (**b**) Petri dish with *Chaetomium* colonies and bacterial species.

**Figure 25 plants-13-00786-f025:**
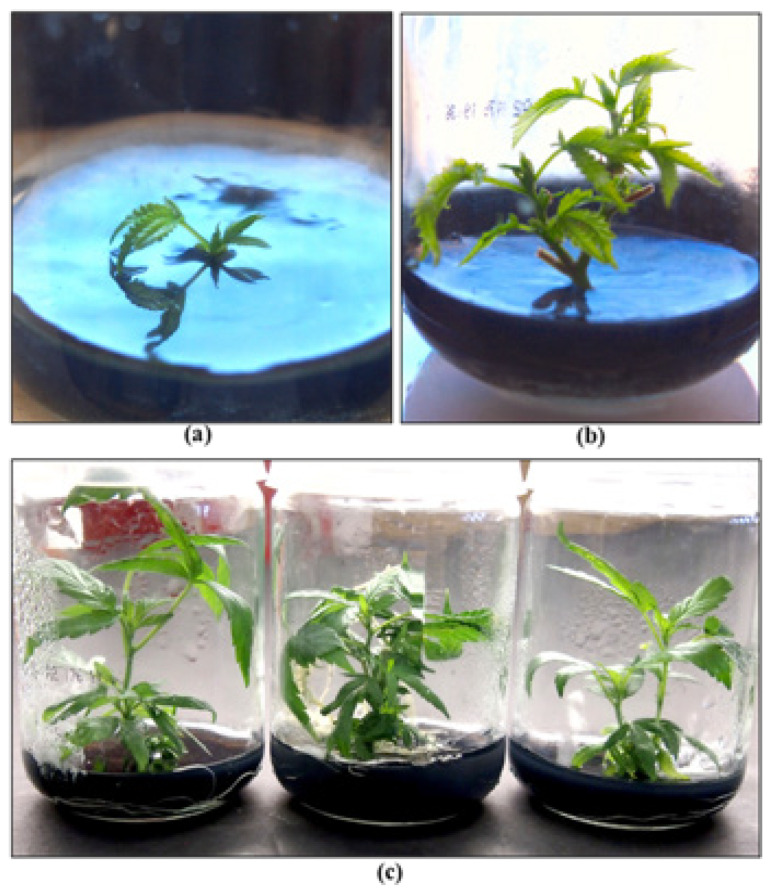
Tissue culture-derived plants of cannabis can be obtained from meristem tips (**a**) and nodal explants (**b**), resulting in growth of a number of genotypes (**c**). The feasibility of generating large-scale production of pathogen-free planting materials awaits further research and development.

**Figure 26 plants-13-00786-f026:**
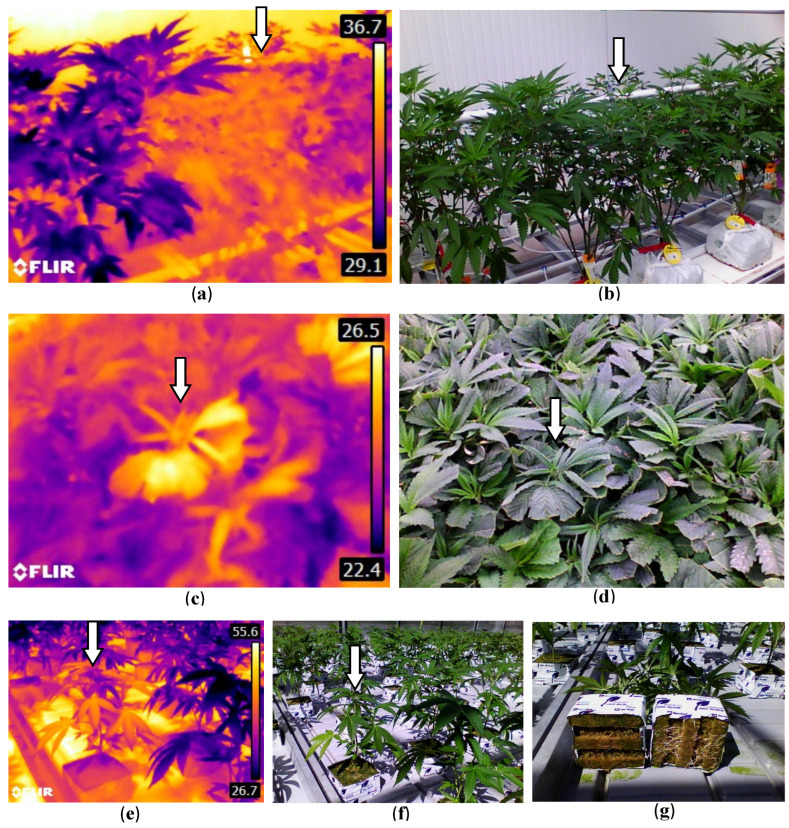
Infrared and digital image comparisons to illustrate changes in plant surface temperatures at different stages of cannabis propagation. (**a**,**b**) A stock plant (arrow) exhibiting a low transpiration rate (and high temperature, in yellow) compared to an adjacent plant with high transpiration (and lower temperature, in purple) shows a difference in surface temperatures that was attributed to infection by a root pathogen. (**c**,**d**) A cutting in the centre of a tray (arrow) with low transpiration (in yellow) surrounded by cuttings with higher transpiration rates. While the former cutting showed no obvious visual symptoms (**d**), early signs of pathogen infection and reduced rooting were observed. (**e**,**f**) A vegetative plant (arrow) with low transpiration (seen in yellow), among other plants with higher transpiration rates, shows notable differences in root health (**g**).

**Table 1 plants-13-00786-t001:** Summary of IDM strategies for four important pathogens affecting cannabis plants.

	HLVd Stunting Disease	*Fusarium*/*Pythium* Root and Crown Rot	*Botrytis cinerea* Bud Rot	Powdery Mildew
Prevention	Test propagative materials and stock plants; utilize pathogen-free planting materials.	Test propagative materials and stock plants; utilize pathogen-free planting materials.	Reduce canopy humidity by adjusting planting density and enhancing air circulation.	Maintain an even climate above 21 °C and vaporize sulfur nightly.
Sanitation	Clean equipment and bench surfaces; destroy diseased plants.	Clean equipment and bench surfaces; actively remove dead or diseased tissues.	Fog growing environment with reduced-risk products prior to planting.	Fog growing environment with reduced-risk products prior to planting.
Protection	Isolate propagative materials and stock plants in controlled access areas.	Apply *Trichoderma harzianum* and *Gliocladium cantenulatum* as a drench to rooted cuttings and plants.	Apply Rootshield HC^®^ on developing inflorescences from day 14 to day 28 of flowering.	Preventatively spray reduced-risk products, such as Suffoil-X and Regalia Maxx, on susceptible genotypes.
Monitoring	Scout regularly for symptoms; routinely sample water and suspect plants.	Scout regularly for symptoms; routinely sample water and suspect plants.	Conduct daily scouting for bud rot from the sixth week of flowering onwards.	Conduct weekly scouting at all plant development stages.
Eradication	Immediately remove and safely dispose of diseased plants at all stages of growth.	Immediately remove and safely dispose of diseased plants at all stages of growth.	Remove and dispose of infected inflorescences; perform post-drying bud rot severity checks.	Remove infected leaves and perform targeted sprays with reduced-risk products.
Genotype Selection	Avoid highly susceptible genotypes; evaluate tolerant genotypes.	Avoid highly susceptible genotypes; evaluate tolerant genotypes.	Avoid planting highly susceptible genotypes during *B. cinerea*-prone periods; evaluate tolerant genotypes.	Avoid highly susceptible genotypes; evaluate tolerant genotypes.

## Data Availability

There were no new data created in this study.

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
