# Peer review of "Integrated Management of Pathogens and Microbes in Cannabis sativa L. (Cannabis) under Greenhouse Conditions"

_plants, 2024, doi:10.3390/plants13060786_

Round 1
Reviewer 1 Report
Comments and Suggestions for Authors
Review of the manuscript entitled " Integrated Management of Pathogens and Microbes on Cannabis sativa L. (Cannabis) under Greenhouse Conditions " by Buirs and Punja.
I appreciate the research presented in the manuscript. It is intriguing and valuable for the scientific community and practice. The authors have provided a thorough analysis supported by numerous tables, figures, diagrams, and photographs. However, the omission of the topic of mycotoxin formation by Fusarium, Aspergillus, Penicillium and Alternata fungi is unfortunate, as it has significant implications for the quality of raw plant material. I suggest that this topic needs to be addressed thoroughly. It is also unclear which methods the authors recommend for identifying the presence of fungi. The description of new and alternative technologies for cannabis disease detection should include Electronic Nose technology. This method has proven effective in detecting Fusarium infections by identifying volatile compounds, making it a useful tool for greenhouse conditions. Additionally, there is evidence that it can detect both infections and mycotoxins. I recommend that the paper be accepted for publication with major revision. The inaccuracies indicated below should be corrected.
1. P7. 2.2.1. Biosecurity and Quarantine Inspection. I would very much like to supplement the analysis with an indication of the diagnostic methods used to monitor the material for fungal pathogens and viruses.
2. P7. 2.2.2. The original description lacks specific values for temperature and humidity that encourage the occurrence of diseases.
3. Rozdział 2.2.4. Sanitary Practices The chapter should be moved and come before the chapter: Testing for Pathogen Presence and Eradication.
4. Fig. 7. Write the names of the fungi in Latin italics.
5. 2.2.5. Utilizing Disease Tolerant Genotypes: fall the Latin names of the fungi causing: powdery mildew and Alternaria leaf blight and bud rot.
6. P.10. Testing for pathogen presence and sanitation are important IDM approaches during plant propagation in greenhouse crops….. Please list which tests these are and briefly characterise them.
7. Fig.10 F. Please give the name of the fungus. Powdery mildew is the name of the disease. The fungus forms spores and not the disease.
8. 2.4.2 Application of Biological Control Agents. What are the advantages and disadvantages of the biological method? What conditions must be met for the successful introduction of BCAs?
9. 2.6.1. Cultural and Environmental Management. It is worth pointing out that UVC lamps should be powerful enough and cover the entire greenhouse, as UVC radiation can destroy fungal spores that are a source of secondary infections.
10. 2.7. Flowering Stage When discussing fungal species found on inflorescences, it's important to note that Penicillium, Alternaria, Fusarium and Aspergillus fungi create mycotoxins that may affect the quality of the plant material.
11. Fig. 15. The pictures do not display consistent fungal cultures, but they are labelled from A to F. It's unclear which fungus is present in each photo. However, in photo 15d, one of the cultures may be Aspergillus spp., and another one is Penicillium or Cladosporium. It would be helpful to add arrows to the photos to assist in identifying the fungal cultures.
12. Fig 22 d, e i f were not quoted in the text.
13. P. orientalis expand the abbreviation because the following genera have been described previously: Pseudomonas, Paecilomyces and Penicillium and ultimately it is not known what species is being described.
14. . Alternative Technologies for Cannabis Disease Detection. Can Electronic Nose technology be used to detect diseases? Many scientific papers show the effectiveness of Fusarium detection using this tool.
Author Response
Please see the attached file that describe the response to reviewer comments

Reviewer 2 Report
Comments and Suggestions for Authors
In this manuscript (plants-2881688) entitled "Integrated Management of Pathogens and Microbes on Cannabis sativa L. (cannabis) Under Greenhouse Conditions" submitted to Plants, Liam Buirs and Zamir K. Punja have discussed combined use of promising strategies for integrated disease management on cannabis plants during greenhouse production. This review is interesting and well-written, but minor points need to be addressed to improve the quality of this manuscript.
1. For the Figure 2, Figure 4, Figure 5, Figure 8, Figure 9, Figure 10, Figure 11, Figure 12, Figure 13, Figure 14, Figure 15, Figure 20, Figure 21, Figure 22, Figure 24, Figure 25, and Figure 26, scale bars should be included in these revised Figures.
2. For the Figure 4, Figure 5, Figure 8, Figure 11, and Figure 14, resistant symptoms should be included as controls in these revised Figures.
3. For the Figure 4, Figure 5, Figure 8, Figure 9, Figure 11, Figure 13, Figure 14, and Figure 21, and Figure 22, pathogen infected area should be indicated with arrows or circles in these revised Figures.
4. For the Figure 7a and 7b, these pictures should be magnified to show more details in the revision. In addition, concentrations of individual products should be labelled in these revised Figures.
Author Response

(The authors gave the same response as above.)

Round 2
Reviewer 1 Report
Comments and Suggestions for Authors
The work has been revised and all the comments have been addressed positively. Congratulations to the authors.